# Information flow, cell types and stereotypy in a full olfactory connectome

Philipp Schlegel[1,2†], Alexander Shakeel Bates[1†], Tomke Stürner[2‡],
Sridhar R Jagannathan[2‡], Nikolas Drummond[2], Joseph Hsu[2,3],
Laia Serratosa Capdevila[2], Alexandre Javier[2], Elizabeth C Marin[2],
Asa Barth-Maron[4], Imaan FM Tamimi[2], Feng Li[3], Gerald M Rubin[3],
Stephen M Plaza[3], Marta Costa[2], Gregory S X E Jefferis[1,2]*

[1]Neurobiology Division, MRC Laboratory of Molecular Biology, Cambridge, United Kingdom; [2]Department of Zoology, University of Cambridge, Cambridge, United Kingdom; [3]Janelia Research Campus, Howard Hughes Medical Institute, Ashburn, United States; [4]Department of Neurobiology, Harvard Medical School, Boston, United States

**Abstract** The *hemibrain* connectome provides large-scale connectivity and morphology information for the majority of the central brain of *Drosophila melanogaster*. Using this data set, we provide a complete description of the *Drosophila* olfactory system, covering all first, second and lateral horn-associated third-order neurons. We develop a generally applicable strategy to extract information flow and layered organisation from connectome graphs, mapping olfactory input to descending interneurons. This identifies a range of motifs including highly lateralised circuits in the antennal lobe and patterns of convergence downstream of the mushroom body and lateral horn. Leveraging a second data set we provide a first quantitative assessment of inter- versus intra-individual stereotypy. Comparing neurons across two brains (three hemispheres) reveals striking similarity in neuronal morphology across brains. Connectivity correlates with morphology and neurons of the same morphological type show similar connection variability within the same brain as across two brains.

*For correspondence:
jefferis@mrc-lmb.cam.ac.uk

†These authors contributed equally to this work
‡These authors also contributed equally to this work

Competing interest: The authors declare that no competing interests exist.

## Introduction

By providing a full account of neurons and networks at synaptic resolution, connectomics can form and inform testable hypotheses for nervous system function. This approach is most powerful when applied at a whole-brain scale. However, until very recently, the handful of whole-brain connectomics data sets have either been restricted to complete nervous systems of a few hundred neurons (i.e. nematode worm; *White et al., 1986*) and *Ciona* tadpole (*Ryan et al., 2016*) or to the sparse tracing of specific circuits, as in larval and adult *Drosophila* (*Zheng et al., 2018*; *Ohyama et al., 2015*).

Now, for the first time, it has become possible to analyse complete connectomes at the scale of the adult vinegar fly, *Drosophila melanogaster*. The 'hemibrain' EM data set (*Scheffer et al., 2020*) provides a step-change in both scale and accessibility: dense reconstruction of roughly 25,000 neurons and 20M synapses comprising approximately half of the central brain of the adult fly. The challenge now lies in extracting meaning from this vast amount of data. In this work, we develop new software, analytical tools and integration strategies, and apply them to annotate and analyse a full sensory connectome.

The fly olfactory system is the largest central brain system that spans first-order sensory neurons to descending premotor neurons; it is a powerful model for the study of sensory processing, learning and memory, and circuit development (*Amin and Lin, 2019*; *Groschner and Miesenböck, 2019*). In this study, we take a principled approach to identify both large-scale information flow and discrete

connectivity motifs using the densely reconstructed hemibrain data set. In addition, we compare and validate results using a second EM data set, the full adult fly brain (FAFB, *Zheng et al., 2018*), which has been used until now for sparse manual circuit tracing (e.g. *Dolan et al., 2018*; *Dolan et al., 2019*; *Sayin et al., 2019*; *Felsenberg et al., 2018*; *Huoviala et al., 2018*; *Zheng et al., 2020*; *Marin et al., 2020*; *Bates et al., 2020b*; *Otto et al., 2020*; *Coates et al., 2020*).

We catalogue first-order receptor neurons innervating the antennal lobe, second-order neurons including all local interneurons and a full survey of third-order olfactory neurons (TOONS; excepting the mushroom body [MB], see *Li et al., 2020*). This classification defines cell types and associates all olfactory neurons with extant functional knowledge in the literature, including the molecular identity of the olfactory information they receive. To further aid human investigation and reasoning in the data set, we develop a computational strategy to classify all olfactory neurons into layers based on their distance from the sensory periphery. We apply this across the full data set, for example, identifying those descending neurons (DNs) (connecting the brain to the ventral nerve cord) that are particularly early targets of the olfactory system.

We also carry out focused analysis at different levels, including the antennal lobe, crucial for initial sensory processing (*Wilson, 2013*), where we reveal highly lateralised microcircuits. After the antennal lobe, information diverges onto two higher olfactory centres, the MB (required for learning) and the lateral horn (LH) (*Vosshall and Stocker, 2007*; *Heisenberg, 2003*; *Grabe and Sachse, 2018*). We analyse reconvergence downstream of these divergent projections as recent evidence suggests that this is crucial to the expression of learned behaviour (*Dolan et al., 2018*; *Dolan et al., 2019*; *Bates et al., 2020b*; *Eschbach et al., 2020*; *Kadow, 2019*).

Finally, building on our recent analysis of second-order olfactory projection neurons in the FAFB data set (*Bates et al., 2020b*), we investigate the stereotypy of cell types and connectivity both within and across brains for select circuits. We show that in two separate cases variability across different brains is similar to variability across the two hemispheres of the same brain. This has important practical implications for the interpretation of connectomics data but also represents a first quantitative effort to understand the individuality of brain connectomes at this scale.

## Results

### Neurons of the olfactory system

The Janelia hemibrain data set comprises most of the right hemisphere of the central brain of an adult female fly and contains ~25,000 largely complete neurons; neurons were automatically segmented and then proofread by humans recovering on average ~39% of their synaptic connectivity (*Scheffer et al., 2020*). Here we process this data into a graph encompassing 12.6M chemical synapses across 1.7M edges (connections) between 24.6k neurons (see Materials and methods). Leveraging this enormous amount of data represents a major challenge. One way to start understanding these data is to group neurons into broad classes and discrete cell types; this enables summaries of large-scale connectivity patterns as well as linking neurons to extant anatomical, physiological and behavioural data.

As a first step, we carried out a comprehensive annotation of all first-, second- and third-order olfactory neurons as well as many higher-order neurons. In particular, we annotate antennal lobe olfactory and thermo/hygrosensory receptor neurons (ALRNs), uni- and multiglomerular projection neurons (uPNs, mPNs), antennal lobe local neurons (ALLNs), lateral horn neurons (LHNs) and lateral horn centrifugal neurons (LHCENT) (*Figure 1*). Defining cell-type annotations depended on a range of computational tools as well as expert review and curation. Broadly, we used NBLAST (*Costa et al., 2016*) to cluster neurons into morphological groups and cross-reference them with existing light-level data and in many cases confirmed typing by comparison with the FAFB EM data set (*Zheng et al., 2018*; *Dorkenwald et al., 2020*).

Our annotation efforts – amounting to 4732 cells and 966 types – were coordinated with those of Kei Ito, Masayoshi Ito and Shin-ya Takemura, who carried out cell typing across the entire hemibrain EM data set (*Scheffer et al., 2020*). Other typing efforts are reported in detail elsewhere (e.g. see *Li et al., 2020* for Kenyon cells [KCs]; mushroom body output neurons [MBONs]; dopaminergic neurons [DANs]; *Hulse et al., 2020* for neurons of the central complex [CXN]) (*Figure 2A B*). All cell-type

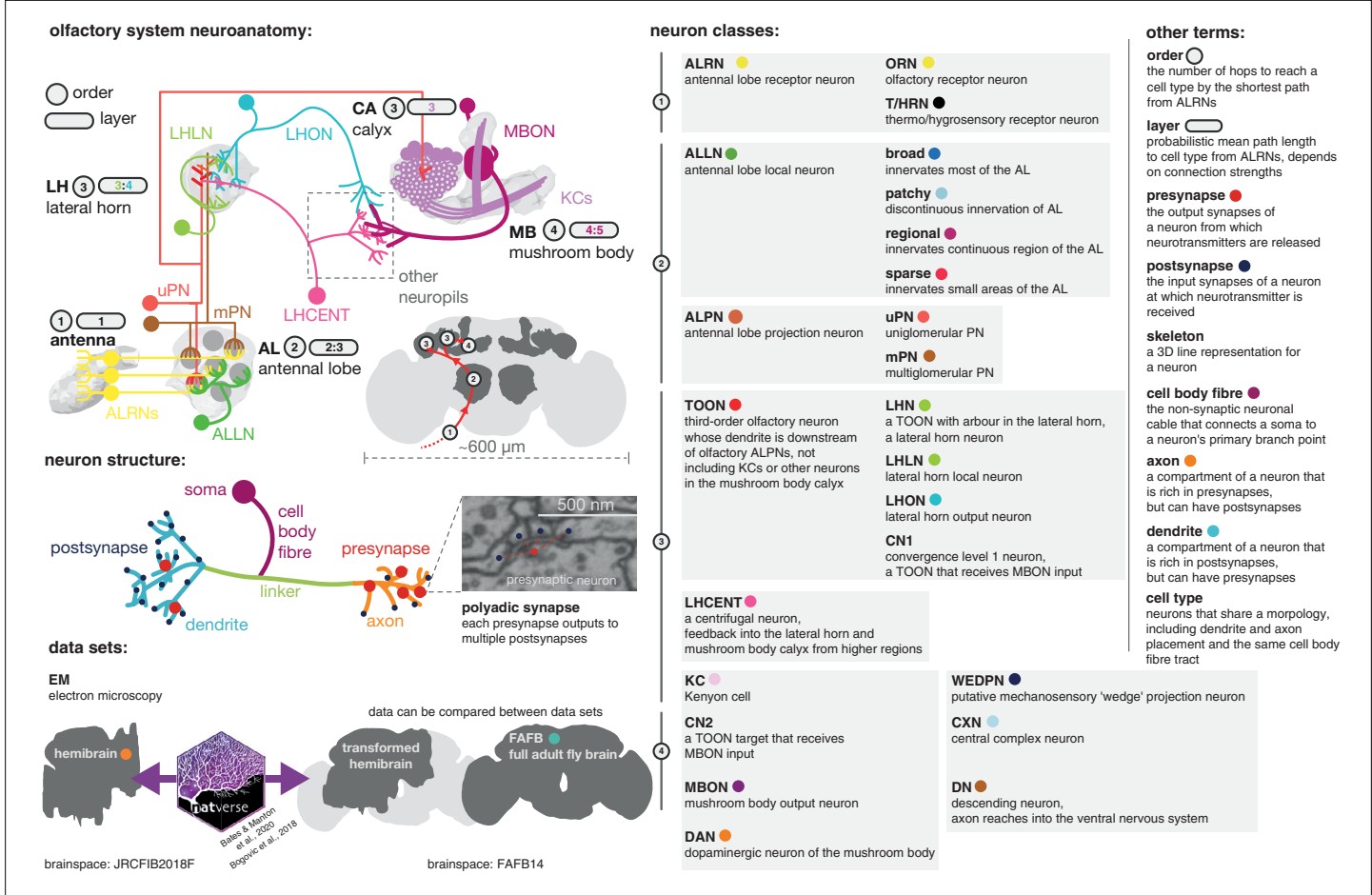

**Figure 1.** Graphical olfactory neuroanatomy glossary. Top left: schematic of the *D. melanogaster* olfactory system showing all its major neuron classes. The 'order' of each neuropil is given in a grey circle, its average layers in a grey lozenge. Inset: the fly brain with a scale bar and early olfactory neuropils shown. Red path is the major feedforward course of olfactory information through the brain. Middle left: a neuron with its compartments is shown. Bottom left: the two EM data sets that feature in this work, the partial dense connectome, the hemibrain and a sparsely reconstructed data set, full adult fly brain (FAFB). Neuroanatomical data can be moved between the two spaces using a bridging registration (*Bogovic et al., 2020*; *Bates et al., 2020a*). Right: major neuron class acronyms are defined. Other neuroanatomical terms are also defined. Coloured dots indicate the colour used to signal these terms in the following figures.

The online version of this article includes the following video for figure 1:

**Figure 1—video 1.** Video of neurons typed in this study grouped by broad class.

https://elifesciences.org/articles/66018/figures#fig1video1

annotations agreed upon by this consortium have already been made available through the hemibrain v1.1 data release at neuprint.janelia.org in May 2020 (*Scheffer et al., 2020*; *Clements et al., 2020*).

Owing to the truncated nature of the hemibrain EM volume, DNs are particularly hard to identify with certainty. By careful review and comparison with other data sets including the full brain FAFB data set, we identified 236 additional DNs beyond the 109 reported in the hemibrain v1.1 release (see Materials and methods and Supplemental data).

## Layers in the olfactory system

Having defined cell types of the olfactory system, a second approach to obtain a system-wide understanding of olfactory organisation is to characterise the connectome graph with respect to an inferred sensory-integrative-motor hierarchy. While this cannot model all aspects of brain function, it provides a human-intelligible summary of information flow.

The basic organisation of the early fly olfactory system is well documented and can be summarised as follows: first-order receptor neurons (ALRNs) in the antennae project to the brain where they

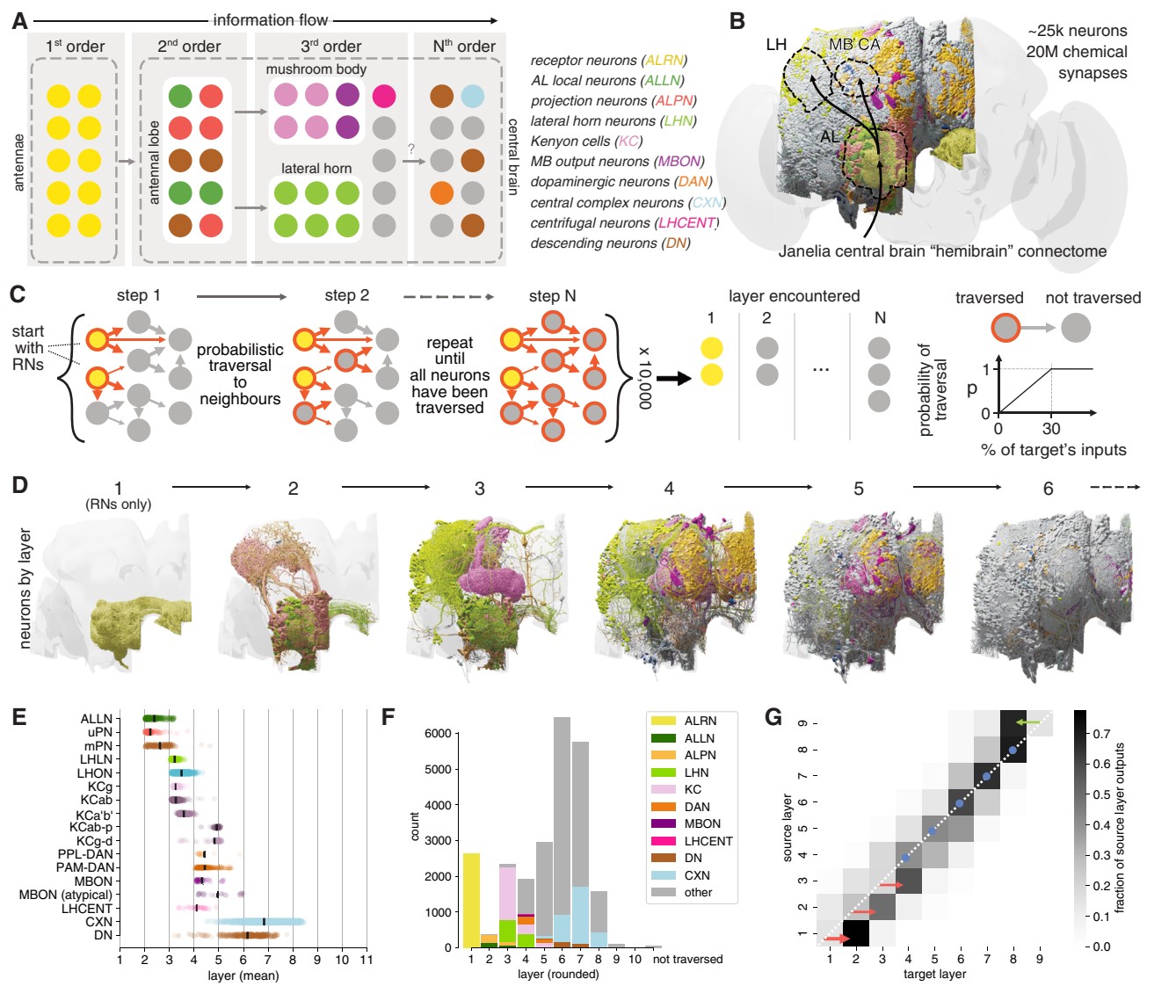

**Figure 2.** Identification of layers in the olfactory system. (**A**) Schematic of the fly's olfactory system. Colours reused in subsequent panels. (**B**) The Janelia Research Campus FlyEM hemibrain connectome. Principal olfactory neuropils as overlay; full brain plotted for reference. (**C**) Graph traversal model used to assign layers to individual neurons. (**D**) Neurons found in the first six layers. (**E**) Mean layer of individual neurons. Black line represents mean across a given neuron class. (**F**) Composition of each layer. (**G**) Connections between layers. AL: antennal lobe; CA: calyx; LH: lateral horn; MB: mushroom body; WEDPN: wedge; ALPN: antennal lobe projection neuron; uPN/mPN: uni-/multiglomerular ALPN.

The online version of this article includes the following video and figure supplement(s) for figure 2:

**Figure supplement 1.** Graph traversal model extended data.

**Figure supplement 2.** Olfactory vs. thermo/hygrosensory layers.

**Figure 2—video 1.** Video of neurons of the first 5 olfactory layers.

https://elifesciences.org/articles/66018/figures#fig2video1

terminate in the antennal lobes and connect to second-order local (ALLNs) and projection neurons (ALPNs). Information is then relayed to TOONs mainly in the MB and the LH (*Figure 2A*; *Wilson, 2013*; *Bates et al., 2020b*). This coarse ordering of first-, second- and third-order neurons is helpful for neuroscientists, but is an oversimplification that has not yet been derived from quantitative analysis. The recent hemibrain dense connectome covers nearly all (known) olfactory neurons; we can therefore for the first time take a systematic approach to layering in this sensory system (*Figure 2B*; *Scheffer et al., 2020*). Here, we employ a simple probabilistic graph traversal model to 'step' through the

olfactory system and record the position at which a given neuron is encountered. We call the positions established by this procedure 'layers' to disambiguate them from the well-established term 'orders' used above. Conceptually, layers correspond to the mean path length from the sensory periphery to any neuron in our graph while taking account of connection strengths; a corresponding quantitative definition of 'orders' would be the shortest path length (which would not consider connection strengths).

In brief, we use the ~2600 ALRNs whose axons terminate in the right antennal lobe as a seed pool (see next section and Materials and methods for details of ALRN identification). The model then traverses to neurons downstream of those in the seed pool in a probabilistic manner: the likelihood of a given neuron being visited increases with the fraction of inputs it receives from neurons in the pool and caps at 30%. For example, a neuron that receives 30%/10%/2% of its synaptic inputs from an ALRN has a 100%/33.3%/0.06% chance to be traversed in the first round. When a neuron is successfully traversed, it is added to the pool and the process is repeated until the entire graph has been traversed. For each neuron, we keep track of at which step it was traversed and use the mean over 10,000 runs to calculate its layer (*Figure 2C*). The probability of traversal is the only free parameter in the model and was tuned empirically using well-known cell types such as uPNs and KCs. While absolute layers depended strongly on this parameterisation, relative layers (e.g. layers of uPNs vs. mPNs) were stable (see Materials and methods and *Figure 2—figure supplement 1A,B* for details).

Running this model on the hemibrain graph set enabled us to assign a layer to ~25,000 neurons (*Figure 2D*). While forgoing many of the complexities of real neural networks such as the sign (i.e. excitation vs inhibition) or types (e.g. axo-dendritic vs. axo-axonic) of connections, it represents a useful simplification to quantitatively define olfactory information flow across the brain, even in deep layers far from the sensory periphery. Practically, these layers also provided a means to validate and refine the naturally iterative process of neuron classification. Early neuron classes are assigned to layers that are intuitively 'correct': for example, most ALPNs and ALLNs appear as expected in the second layer. However, close inspection revealed marked differences, some of which we analyse in-depth in subsequent sections. Initial observations include the fact that mPNs appear, on average, slightly later than their uniglomerular counterparts (*Figure 2E F*). This is likely due to mPNs receiving significant input from other second-order neurons (i.e. uPNs and ALLNs) in addition to their direct input from receptor neurons.

Neurons traditionally seen as third-order components of the two arms of the olfactory system (KCs, in the MB calyx and LHNs) actually span two layers (3 and 4) due to lateral connections. Among the KCs, those with primarily visual inputs (KCαβ-p and KCγ-d) appear later than those with primarily olfactory input.

DNs are few (~350–600/hemisphere) and represent the principal connection to the motor centres in the ventral nervous system (*Hsu and Bhandawat, 2016*; *Namiki et al., 2018*). We find that the majority of DNs are distant from olfactory inputs (sixth layer). However, a small subset appear as third- or fourth-layer neurons. These may represent shortcuts between the olfactory and motor systems used for behaviours that are hard-wired or require fast responses.

In layers 1–3, neurons talk primarily to others in the next higher layer (*Figure 2G*). Layers 4–7 then show increased intra-layer connectivity. At layer 6 the directionality begins to reverse: layers start connecting more strongly to neurons in the same layer and eventually the previous one(s). This may indicate that the flow of information inverts at this point and that layers 6–7 represents the 'deepest' point of the olfactory system.

The above analysis combines olfactory and thermo- and hygrosensory ALRNs (see *Figure 2— figure supplement 2* for a separate break down). We will use these layers as we proceed through the olfactory system, classifying neurons in detail and extracting connectivity motifs.

## Antennal lobe receptor neurons

ALRNs that express the same receptor project to the same globular compartments, glomeruli, of the olfactory bulb in vertebrates (*Su et al., 2009*), or the antennal lobe in insects (*Couto et al., 2005*; *Fishilevich and Vosshall, 2005*; *Vosshall et al., 2000*). In *Drosophila*, ALRNs are either unilateral or (more commonly) bilateral and connect with ALLNs and ALPNs (*Figure 3A*). We identified ~2600 ALRNs in the hemibrain data set as projecting to one of 58 glomeruli of the right antennal lobe by manually curating a list of candidate neurons (*Figure 3B*, see Materials and methods for details). Notably, we

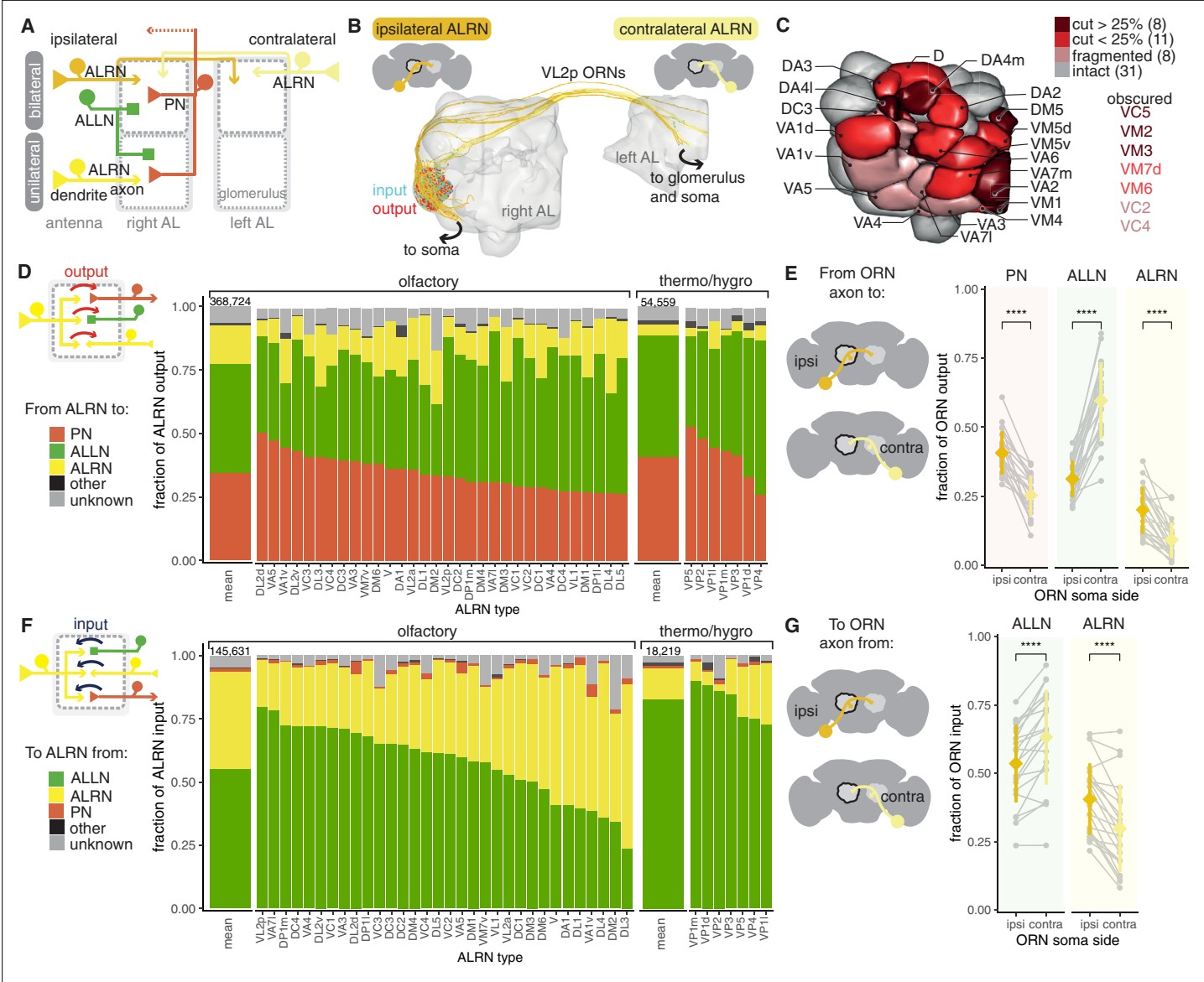

**Figure 3.** Antennal lobe (AL) receptor neurons (ALRNs) mostly target projection and local neurons. (**A**) Summary schematic of antennal lobe ALRN classification and the major cell types present in the antennal lobe that interact with them. ALLN: antennal lobe local neurons; ALPN: antennal lobe projection neuron. (**B**) Ipsilateral and contralateral VL2p olfactory ALRNs (olfactory receptor neurons [ORNs]) in the right antennal lobe. The somas are not visible as they are cut off from the volume. Output synapses in red, input ones in blue. (**C**) Antennal lobe glomerular meshes (generated from ALRNs) showing which glomeruli are truncated and by how much (qualitative assessment). ALRN types in whole glomeruli but with fragmented ALRNs, which prevents assignment of soma side, are also shown. (**D**) Fraction of ALRN output per type. The leftmost bar is the mean for olfactory or thermo/hygrosensory ALRNs, with number of synapses on top. (**E**) Fraction of ipsilateral (ipsi) or contralateral (contra) ORN output to ALLNs, ALPNs and ALRNs. Means were compared using Wilcoxon two-sample tests. (**F**) Fraction of ALRN input per type. The leftmost bar is the mean for ORNs and thermo-receptor neurons (TRNs)/hygro-receptor neurons (HRNs), with number of synapses on top. (**G**) Fraction of ipsilateral (ipsi) or contralateral (contra) ORN input from ALLNs and ALRNs. Means were compared using Wilcoxon two-sample tests. Significance values: ns: p>0.05; *p≤0.05; **p≤0.01; ***p≤0.001; ****p≤0.0001.

The online version of this article includes the following figure supplement(s) for figure 3:

**Figure supplement 1.** Annotation of antennal lobe receptor neuron (ALRN) bodies and connectivity features.

**Figure supplement 2.** Antennal lobe receptor neuron (ALRN) clustering and subdivision of the VM6 glomerulus.

renamed three glomeruli to resolve conflicting information in past literature: VC5 → VM6, VC3m → VC5, VC3l → VC3 (see Materials and methods for details). These changes will appear in version 1.3 of the hemibrain data set and have been coordinated with other research groups working on these glomeruli (*Task et al., 2020*; *Vulpe et al., 2021*).

19 glomeruli are either medially or anteriorly truncated in the hemibrain volume, while an additional 8 glomeruli are intact but have very fragmented ALRNs. This affects our recovery and identification of ALRNs (*Figure 3—figure supplement 1*), and we estimate our coverage per glomerulus to be on average around 70% compared to previously published counts (*Rybak et al., 2016*; *Tobin and Wilson, 2017*; *Horne et al., 2018*; *Stocker, 2001*; *Grabe et al., 2016*; *Figure 3—figure supplement 1*). In subsequent analysis, ALRNs of truncated glomeruli are not included. The 31 fully intact glomeruli include all the thermo- and hygrosensory ones (n = 7) and 24 olfactory ones (*Figure 3C*; *Marin et al., 2020*; *Bates et al., 2020b*). Thermo- and hygrosensory ALRNs (TRN/HRNs) are mostly unilateral (6/7) with 8 ALRNs per type on average, while olfactory ALRNs (olfactory receptor neurons [ORNs]) are predominantly bilateral (22/24) with 27 ALRNs per type (*Figure 3—figure supplement 1*).

Building on our comprehensive analysis of ALRNs, we have now found that ALRNs of the VM6 glomerulus consist of three anatomically distinct subpopulations (VM6v, VM6m and VM6l) connecting to the same postsynaptic PNs; these populations differ in their receptor expression and their origin in peripheral sense organs (*Task et al., 2020*; *Vulpe et al., 2021*). These findings helped to explain previous uncertainties about this part of the antennal lobe, which have resulted in many nomenclature discrepancies in the prior literature. Having an almost full set of ALRNs in the hemibrain, we asked whether any other glomerulus showed a similar subdivision. Based on morphological clustering, we can confirm the VM6 subpartition but also conclude that none of the other glomeruli exhibit a similar potential for further partitioning (*Figure 3—figure supplement 2*,B) Moreover, we find that the VM6 ALRN subpopulations, while morphologically distinct, appear to converge onto the same downstream targets. None of the uniglomerular ALPNs show a clear preference towards any individual VM6 ALRN subtype. Likewise, a clustering of VM6 ALRNs based on their downstream connectivity does not align with the morphology-based clustering (*Figure 3—figure supplement 2*).

Besides providing the first large-scale quantification of synaptic connectivity in the adult antennal lobe, we focused on two specific aspects: first, connection differences between the olfactory and thermo/hygrosensory ALRNs; second, wiring differences between ALRNs originating from the ipsilateral and contralateral antennae. Most of the output from ALRNs is to ALLNs (43% and 48% from ORNs and TRN/HRNs, respectively), followed by ALPNs (34% and 41% from ORNs and TRNs/HRNs, respectively). The remainder is either accounted for by ALRN-ALRN connectivity or other targets that are not ALRNs, ALPNs or ALLNs. This connectivity profile is similar to what has been reported for the larva (*Berck et al., 2016*) even though the number of neurons and types has increased significantly (*Scheffer et al., 2020*; *Bates et al., 2020b*). They are also consistent with two previous studies of single glomeruli in the adult fly (*Horne et al., 2018*; *Tobin and Wilson, 2017*).

We find that compared to ORNs TRN/HRNs spend more of their output budget on connections to ALPNs (41% vs. 34%), and this difference seems to be mostly accounted for by the very low level of axo-axonic TRN/HRN to ALRN connectivity (*Figure 3D*). Type specificity is also clearly apparent, however, with individual ALRN types showing different presynaptic densities (*Figure 3—figure supplement 1*) as well as particular profiles of ALLN and ALPN output (*Figure 3D,E*). Two pheromone-sensitive ORN types, DA1 and VA1v, output the most to other neurons. Their main target are the AL-AST1 neurons which arbourise in and receive input from a subset of antennal lobe glomeruli and output mostly in the antennal lobe and the saddle, a region that includes the antennal mechanosensory and motor centre (AMMC) (*Scheffer et al., 2020*; *Tanaka et al., 2012a*).

The majority of input onto ALRNs is from ALLNs and other ALRNs, and can vary widely – in particular across the different ORN types (*Figure 3F*). Connections between ALRNs occur almost exclusively between neurons of the same type, for example, DA1→DA1 but not DA1→DA2 (data not shown). This is consistent with previous reports of connections between axon terminals of gustatory or mechanosensory neurons in larval and adult *Drosophila* (*Hampel et al., 2020*; *Miroschnikow et al., 2018*). The functional relevance of these connections is unclear. In contrast, ALLNs have been shown to regulate and coordinate activity across glomeruli via lateral inhibition (e.g. see *Mohamed et al., 2019*; *Wilson and Laurent, 2005*) and ALLN→ALRN connections likely play a role. We find that pheromone-sensitive ORNs (targeting DA1, DL3 and VA1v) are amongst those with the least ALLN input onto their

terminals, suggesting that they might be less strongly modulated by other channels. As expected from analysis of output connectivity, TRNs and HRNs mostly receive input from ALLNs.

Breaking down bilateral ORN connectivity by laterality highlights a distinct behaviour of ALLNs: on average, contralateral ORNs provide more information to, and receive more information from, ALLNs than ipsilateral ORNs (*Figure 3E G* and *Figure 3—figure supplement 1*). This is in contrast to ALPNs, whose behaviour is consistent with previous reports (*Gaudry et al., 2013*; *Agarwal and Isacoff, 2011*). This bias could help the animal to respond to lateralised odour sources.

## Antennal lobe local neurons

Light microscopy studies have estimated ~200 ALLNs (*Chou et al., 2010*). ALLNs have complex inhibitory or excitatory synaptic interactions with all other neuron types in the antennal lobe, that is, the dendrites of outgoing ALPNs, the axons of incoming ALRNs and other ALLNs. In particular, ALLN-ALLN connections are thought to facilitate communication across glomeruli, implementing gain control for fine-tuning of olfactory behaviour (*Root et al., 2008*; *Olsen and Wilson, 2008*). ALLNs are diverse in morphology, connectivity, firing patterns and neurotransmitter profiles and critically, in the adult fly brain, they do not appear to be completely stereotyped between individuals (*Seki et al., 2010*; *Okada et al., 2009*; *Chou et al., 2010*; *Berck et al., 2016*). Previously, six types of ALLNs (LN1-LN6) had been defined mainly based on the expression of specific GAL4 lines (*Tanaka et al., 2012a*). The hemibrain data set now provides us with the first opportunity to identify and analyse a complete set of ALLNs at single-cell resolution.

We find 196 ALLNs in the right hemisphere which we assign to 5 lineages, 4 morphological classes, 25 anatomical groups and 74 cell types (*Figure 4A–D* and *Figure 5—figure supplement 1*). ALLNs derive from three main neuroblast clones: the lateral neuroblast lineage ('l' and 'l2' from ALl1), the ventral neuroblast lineage ('v' from ALv1) and the ventral ALLN-specific lineage ('v2' from ALv2) (*Sen et al., 2014*). Their cell bodies cluster dorsolateral, ventromedial or ventrolateral to the antennal lobe or in the gnathal ganglion (referred to as il3; *Bates et al., 2020b*; *Shang et al., 2007*; *Tanaka et al., 2012a*). Around 40% (78) of the ALLNs are bilateral and also project to the left antennal lobe; most of these (49) originate from the v2 lineage. Correspondingly, we identified fragments of 88 ALLNs that originate in the left and project to the right antennal lobe (*Figure 4A*).

The morphological classification of ALLNs is based on their glomerular innervation patterns reported by *Chou et al., 2010*: 'broad' ALLNs innervate all or most of the AL; 'patchy' ALLNs exhibit characteristic discontinuous innervation; 'regional' ALLNs innervate large continuous regions of the AL; and 'sparse' ALLNs innervate only a small area of the antennal lobe (*Figure 4B*). These differences in innervation patterns can be quantified: for each ALLN we ranked glomeruli by the number of synapses placed inside (descending) and further normalised them per ALLN. Finally we summed those numbers up cumulatively per ALLN. Sparse ALLNs place their synapses in a select few glomeruli (typically <10), while broad ALLNs distribute their synapses evenly across the majority of glomeruli (typically >30) (*Figure 4E*). Anatomical groups are then defined as sets of cell types with similar morphological features.

Previous research has shown that while most ALLNs exhibit input and outputs in all innervated glomeruli, some show signs of polarisation (*Chou et al., 2010*). Indeed, regional and sparse ALLNs can mostly be split into an axonic and a dendritic compartment, while broad and patchy ALLNs tend to be less polarised (*Figure 4F*). Axon-dendrite segregation may facilitate specific inter-glomerular interactions. In particular, looking at the most polarised ALLNs (score >0.1), differential dendritic input and axonic output are apparent with respect to pairs of thermo/hygrosensory glomeruli of opposing valences (*Figure 4—figure supplement 1G*). Significantly, v2LN49 neurons receive dendritic input in the 'heating' glomerulus VP2 (*Ni et al., 2013*) and have axonic outputs in the 'cooling' glomerulus VP3 (*Gallio et al., 2011*; *Budelli et al., 2019*), while l2LN20 and l2LN21 perform the opposite operation. An interesting odour example is lLN17 which receives dendritic inputs from pheromone glomerulus DA1 and has axonic output to another pheromone glomerulus, VA1v (*Kurtovic et al., 2007*; *Dweck et al., 2015*). Such interactions might help regulate female receptivity.

ALLNs principally connect to ALRNs, ALPNs and other ALLNs. Connectivity differs greatly between ALLN cell types, even within groups. Smaller ALLNs (sparse, regional) tend to receive a greater fraction of direct ALRN input than larger ALLNs (broad, patchy) and are therefore assigned to earlier

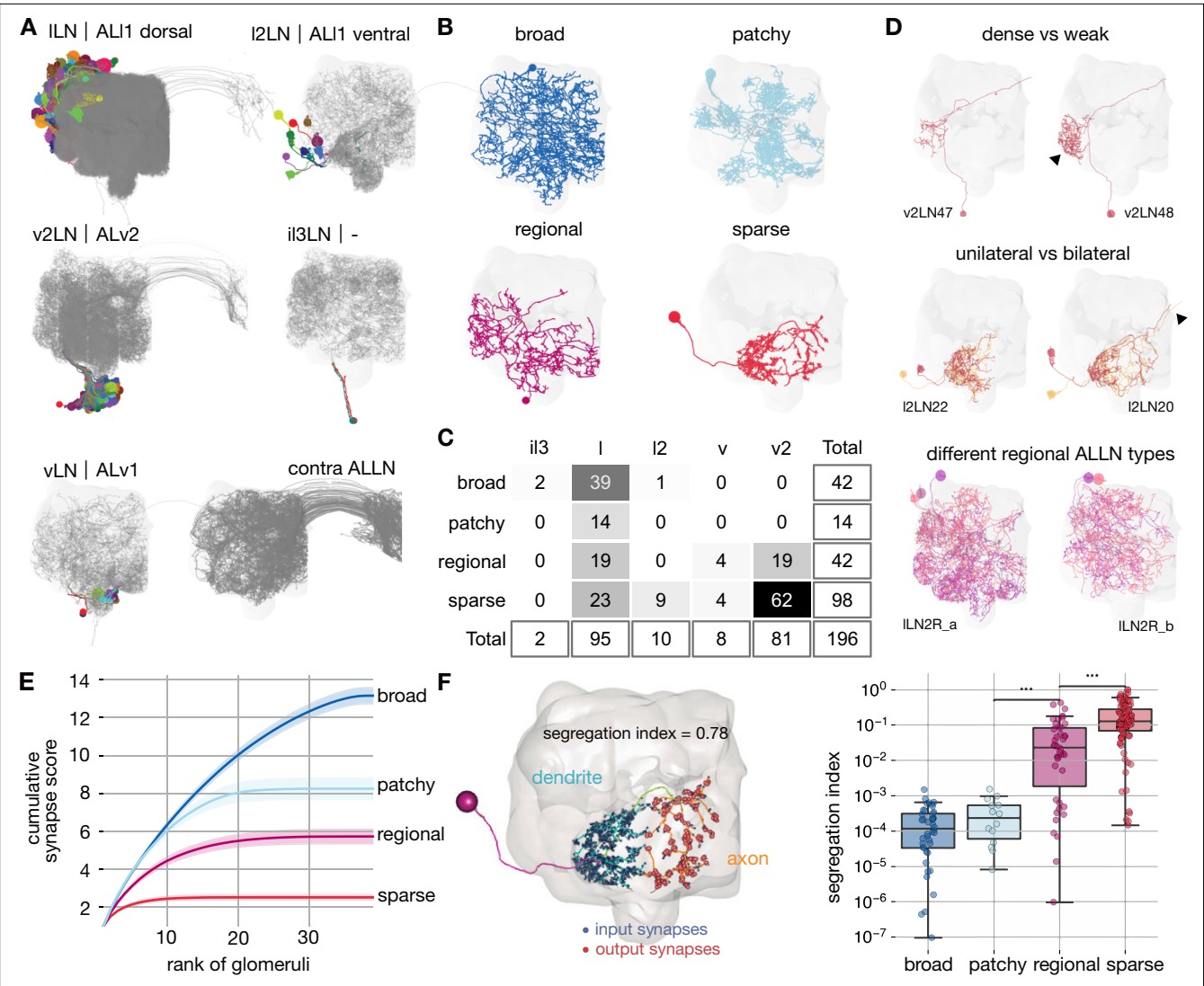

**Figure 4.** Cell typing, morphological classification and polarity of antennal lobe local neurons. (**A**) Antennal lobe local neurons (ALLNs) classified by hemilineage and contralateral ALLNs (contra ALLN), along with the antennal lobe mesh in the background. Soma locations (circles) and primary neurite tracts are illustrated in multicolours. (**B**) Morphological classes of ALLNs. A representative example of each category is shown. (**C**) Number of ALLNs per hemilineage and morphological class. (**D**) Representative examples illustrating criteria used for typing: unilateral and bilateral neurites, lineage identity, area innervated by ALLN neurites and their density. Arrowheads point towards dense innervation and bilateral projection. (**E**) Synapse score per morphological class. Cumulative number of synapses is computed per ranked glomerulus (by number of synapses) and plotted against its rank. Envelopes represent standard error of the mean. (**F**) Polarisation of neurites per morphological class. Segregation index is a metric for how polarised a neuron is; the higher the score the more polarised the neuron (*Schneider-Mizell et al., 2016*). Left inset shows a sparse ALLN, l2LN21, as an example of a highly polarised ALLN. Significance values: *p≤0.05; **p≤0.01; ***p≤0.001; ****p≤0.0001; pairwise Tukey-HSD post-hoc test.

The online version of this article includes the following figure supplement(s) for figure 4:

**Figure supplement 1.** Antennal lobe local neuron (ALLN) glomerular innervation patterns.

layers (*Figure 5*). Strong ALLN-ALLN connectivity arises mostly from the broad ALLNs of the lateral lineage (*Figure 5C*). They may act as master regulators of the ALLN network.

Breaking down the input onto ALLNs, we see that some have very high specificity for specific glomeruli: for example, vLN24 receives 67% of its ALRN input from the $CO_2$ responsive V glomerulus ORNs (*Figure 5B F*). Importantly, we also observe substantial differences in the degree of ipsi-versus contralateral ALRN input across the ALLN population (*Figure 5B*). At one end of the spectrum, regional vLNs receive more than 10 times as much input from ipsilateral versus contralateral ORNs; in contrast, broad il3LNs receive fivefold more contralateral ORN input. These broad il3LNs, a single

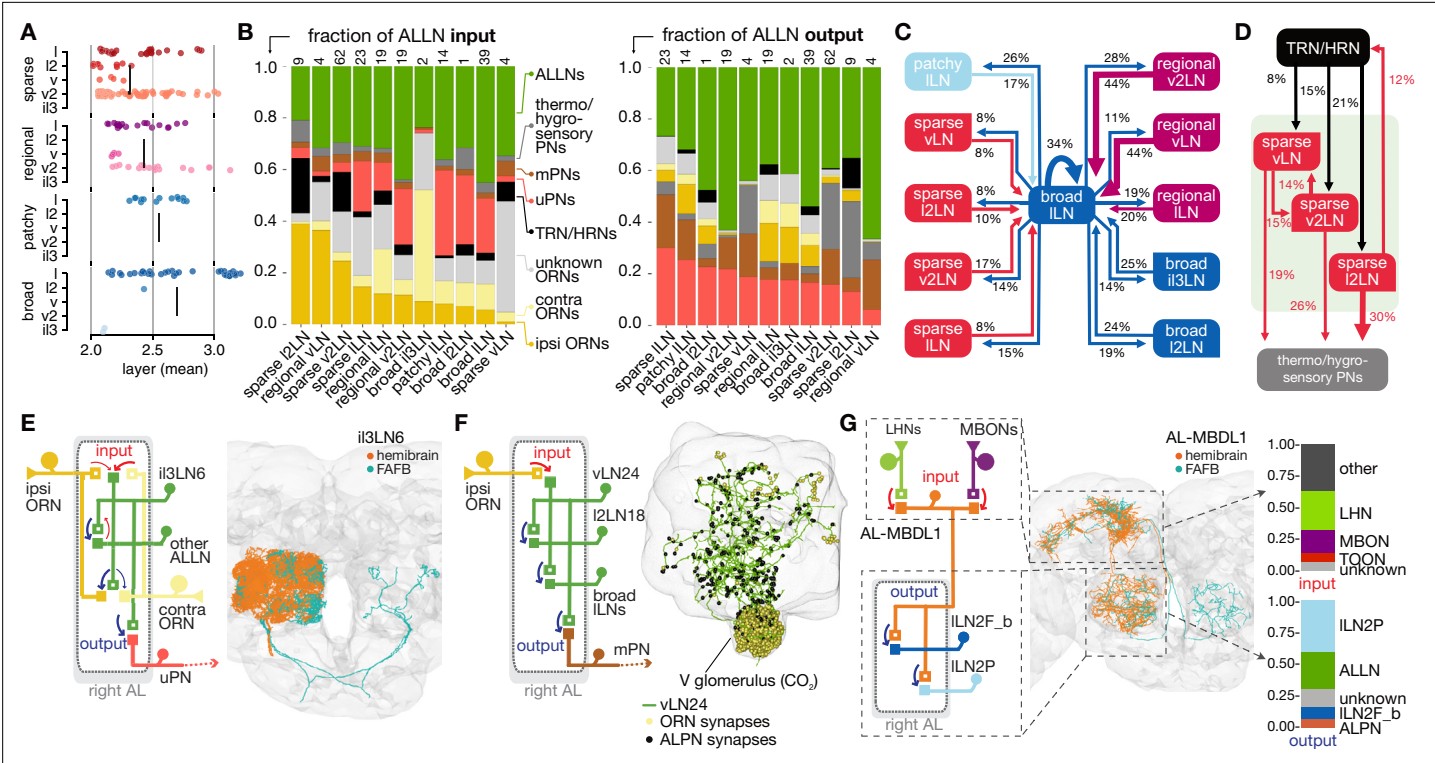

**Figure 5.** Antennal lobe (AL) local neuron (ALLN) connectivity and example circuit motifs. (**A**) Layers of ALLNs. Vertical lines indicate group mean. (**B**) Fraction of ALLN input (left) and output (right) for different ALLN groups. Number of neurons per category is shown at the top of each bar. Where possible olfactory receptor neurons (ORNs) are split into 'ipsi-' and 'contra'-lateral ('unknown' ORNs mostly correspond to those that are fragmented or belonging to truncated glomeruli). (**C**) Diagram illustrating ALLN-ALLN connectivity. ALLN groups are coloured by morphological class. (**D**) Diagram illustrating the most prominent ALLN connections to thermo/hygrosensory antennal lobe receptor neurons (ALRNs) and antennal lobe projection neurons (ALPNs). (**E–G**) Examples of ALLN connectivity. (**E**) A pair of broad ALLNs, il3LN6, cross-matched to the FAFB Keystone ALLNs. (**F**) An example of type regional ALLNs, vLN24, that receives specialised input in the V glomerulus. (**G**) A bilateral medial bundle neuron, AL-MBDL1, that integrates lateral horn neuron (LHN) and mushroom body output neuron (MBON) input and outputs to two specific types of ALLNs, broad lLN2F_b and patchy lLN2P.

The online version of this article includes the following figure supplement(s) for figure 5:

**Figure supplement 1.** Antennal lobe local neuron (ALLN) groups.

pair of bilateral neurons likely analogous to the larval Keystone ALLNs (*Coates et al., 2020*; *Berck et al., 2016*), interact strongly with broad lLNs while also providing strong presynaptic inhibition onto ORN terminals (*Figure 5B, C E*). This may represent a major mechanism by which contralateral odour information influences the ipsilateral ALLN-ALLN network.

Curiously, sparse ALLN cell types receive a large proportion of their input from TRNs/HRNs (sparse l2LNs: 21%; sparse v2LN: 15%; sparse vLNs: 8%). Other cell types receive at most 5%. Indeed, comparing antennal lobe innervation patterns against a random null model suggests that sparse ALLNs are more likely to co-innervate thermo/hygrosensory glomeruli (*Figure 4—figure supplement 1H*). Similarly, when we examine ALPN connectivity, we see that sparse ALLN cell types send a large proportion of their output to THPNs (sparse l2LNs: 30%; sparse v2LNs: 26%; sparse vLNs: 19%). Other cell types receive at most 5%. This indicates that sparse ALLNs may be modulating very specific thermo/hygrosensory information among the circuitry within the AL. In combination, this suggests the existence of a local network made of sparse ALLNs that encompasses only the non-olfactory, thermo/hygrosensory glomeruli.

Regional ALLNs, on the other hand, co-innervate combinations with the DP1m (responds to e.g. 3-hexanone, apple cider vinegar) or DP1l (acetic acid) glomeruli, which may be key food odours detecting glomeruli and are some of the largest in the antennal lobe. The patchy ALLNs' co-innervation does not differ from the null model, which agrees with observations from light-level data (*Chou et al., 2010*).

About half of the ALLNs also feedback strongly onto ALRN axons. Interestingly, ALLNs of lineages v2 and v send very little output to the ALRNs (regional v2LNs: 1.8%; sparse vLNs: 1.7%; sparse v2LNs: 5.2%; sparse l2LNs: 5%; regional vLNs: 1.2%) compared with other ALLNs, which spend >16% of their outputs on ALRN axons. The ALLNs that modulate ALRN axons likely execute circuit functions distinct from those that do not, perhaps operating to quickly adapt and stabilise ALRN responses.

We also observe centrifugal feedback from higher olfactory areas, into the antennal lobe. The antennal lobe-associated median bundle neuron (AL-MBDL1) is a centrifugal modulatory neuron that integrates input from the MB and the LH (*Tanaka et al., 2012a*; *Figure 5G*). It arbourises widely in the antennal lobe and outputs onto two specific sets of ALLNs: the 14 patchy lLN2P and a pair of broad lLN2F_b neurons (*Figure 5F*). This means that the superior brain regions may be able to exercise control over the ALLN-ALLN network through AL-MBDL1 activity.

## Stereotypy in olfactory projection neurons

Glomeruli are innervated by principal cells, mitral and tufted cells in vertebrates and projection neurons (ALPNs) in insects, which convey odour, temperature and humidity information to third-order neurons in higher brain regions (*Figure 6A*). These neurons may be excitatory or inhibitory, and either uniglomerular (uPNs) or multiglomerular (mPNs), that is, sampling from a single glomerulus or multiple glomeruli, respectively (*Bates et al., 2020b*; *Tanaka et al., 2012a*).

Most uPNs are well studied and have been shown to be highly stereotyped (*Jefferis et al., 2007*) which makes cross-matching these cell types relatively straightforward. In particular, the 'canonical' uPN types that have been extensively studied in the past (*Yu et al., 2010*; *Ito et al., 2013*; *Tanaka et al., 2012a*; *Grabe et al., 2016*) are easily and unambiguously identifiable in the hemibrain. The situation is less clear for mPNs, for which there is as yet no conclusive cell typing. mPN types were therefore determined by the aforementioned consortium using a combination of within-data set morphological and connectivity clustering under the assumption that these types would be further refined in future releases. In combination, hemibrain v1.1 features 188 ALPN types.

We previously described the morphology of 164 uPNs (forming 81 different types) and 181 mPNs (untyped) in the right hemisphere in the FAFB EM volume (*Bates et al., 2020b*). Here, we add a third ALPN data set from the left hemisphere of FAFB. Together, these data allow us to assess numerical and morphological stereotypy within (FAFB right vs. left) and across animals (hemibrain vs. FAFB left/right) (*Figure 6A*).

First, we find that the total number of ALPNs is largely consistent across brains as well as across hemispheres of the same brain (*Figure 6B*). For uPN types, we find similar variations in ALPN numbers within and across animals (*Figure 6C*, *Figure 6—figure supplement 1A*). Interestingly, variation only occurs in larval-born 'secondary' neurons but not with embryonic 'primary' neurons, and is more obvious for later-born neurons (*Figure 6—figure supplement 1A*).

To obtain a quantitative assessment of morphological stereotypy, we first transformed all ALPNs into the same template brain space (JRC2018F, *Bogovic et al., 2020*) and mirrored the left FAFB ALPNs onto the right (see *Bates et al., 2020a* and Materials and methods for details). Next, we used NBLAST (*Costa et al., 2016*) to generate pairwise morphological similarity scores across the three sets of ALPNs (*Figure 6D*). Due to the large number of data points (~23k per comparison), the distributions of within- and across-animal scores are statistically different (p<0.05, Kolmogorov-Smirnov test); however, the effect size is extremely small. Importantly, the top within-animal scores are on average not higher than those from the across-animal comparisons. This suggests that neurons are as stereotyped within one brain (i.e. across left/right brain hemispheres) as they are between two brains.

An open question is whether individual cells and cell types can be recovered across animals. For neurons like the canonical uPNs, this has already been shown but it is less clear for, for example, the mPNs. First, for nearly all hemibrain ALPN we find a match in FAFB and for most neurons the top NBLAST hit is already a decent match (data not shown). The few cases without an obvious match are likely due to truncation in the hemibrain or developmental abnormalities of the neuron.

Next, we sought to reproduce hemibrain cell types across data sets. Biological variability might well produce a partition in one animal that is not present in another, and *vice versa* (*Figure 6E*). To address this, we used the top across-data set NBLAST scores to generate 197 clusters of morphologically similar neurons across the three populations of PNs (*Figure 6D–F*; see Materials and methods for details). This is slightly more than the 188 PN types listed for hemibrain v1.1 and might indicate that

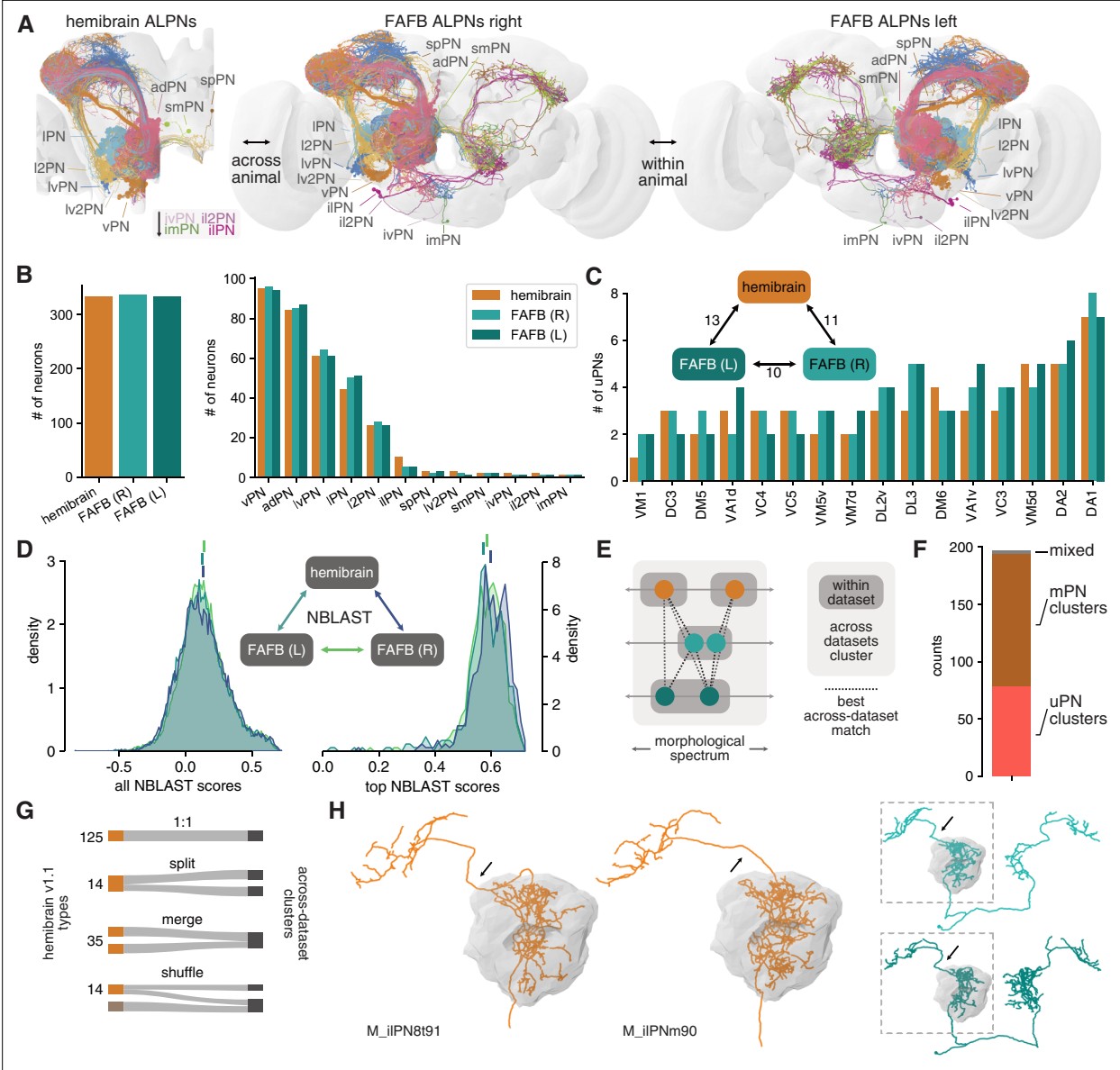

**Figure 6.** Numerical and morphological across – and within-animal stereotypy. (**A**) Antennal lobe projection neurons (ALPNs) reconstructed in the hemibrain and from the left and right hemispheres of the full adult fly brain (FAFB) EM volume. (**B**) Overall ALPN counts are almost identical across hemispheres as well as across animals. (**C**) 17/56 uniglomerular projection neuron (uPN) types show variations in numbers. Numbers in triangle count instances of variation in numbers. (**D**) Across-data set NBLAST similarity scores are much the same. All scores on the left, only pairwise top scores on the right. Top lines represent means. (**E**) Clustering approach based on best across-data set matches. (**F**) Total number of across-data set clusters by composition. (**G**) Quantification of discrepancies between hemibrain v1.1 types and the across-data set clusters. See also *Figure 6—figure supplement 1F*. (**H**) Example where two hemibrain types merge into one across-data set cluster (2). One of the hemibrain neurons takes the 'wrong' antennal lobe tract (arrows) and has therefore been incorrectly given a separate type. See *Figure 6—figure supplement 1G-J* for more examples.

The online version of this article includes the following figure supplement(s) for figure 6:

**Figure supplement 1.** Comparison of antennal lobe projection neurons (ALPNs) across three hemispheres.

our approach over-segments the data. Indeed, the majority of our clusters represent 1:1:N matches (*Figure 6—figure supplement 1B*).

In general, the correspondence between hemibrain types and the across-data set clusters is good: ~74% of hemibrain types map to either one single cluster or split into separate clusters that contain only this cell type (a consequence of the over-segmentation) (*Figure 6G*). 35 (19%) hemibrain types merge into larger clusters. For example, M_ilPNm90 and M_ilPN8t91 were assigned separate types

because of differences in the axonal tract. In comparison with FAFB ALPNs, it becomes apparent that M_ilPNm90's tract is an exception and they indeed belong to the same type (*Figure 6H*). Only 14 (~7%) hemibrain types are shuffled into different clusters. We also note a few instances of discrepancies between classifications of co-clustered neurons which will be solved in future hemibrain/FAFB releases.

In summary, these results are encouraging with respect to matching neurons (types) across data sets while simultaneously illustrating potential pitfalls of cell typing based on a single data set.

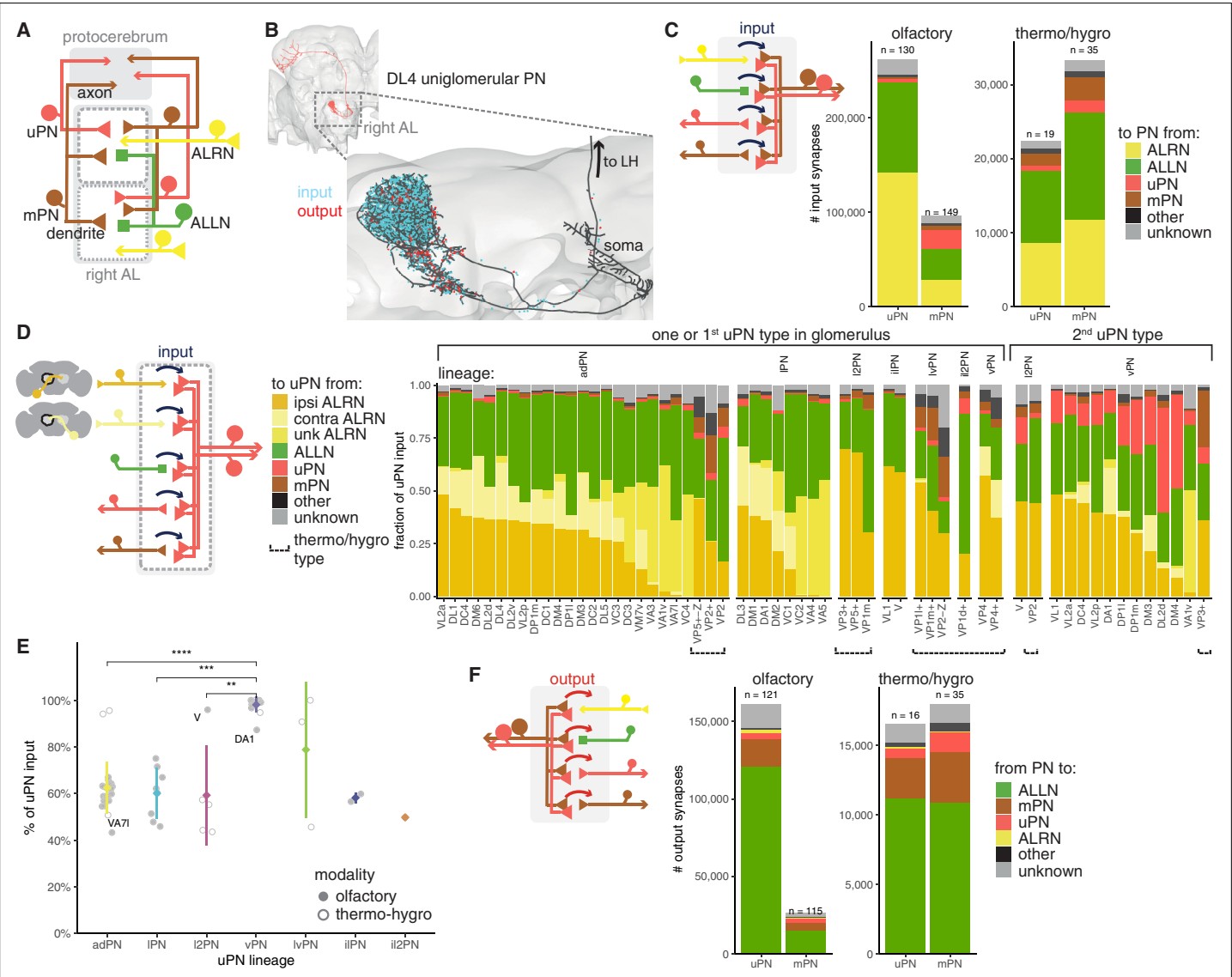

**Figure 7.** Antennal lobe (AL) projection neuron (ALPN) connectivity in the right antennal lobe. (**A**) Summary schematic of ALPN classification and the major cell types present in the antennal lobe that interact with them. uPN: uniglomerular ALPN; mPN: multiglomerular ALPN. (**B**) DL4 uniglomerular PN showing inputs (cyan) and outputs (red). (**C**) Number of input synapses onto olfactory or thermo/hygrosensory uPNs and mPNs. Number of neurons in each category shown at the top of the bar. (**D**) Fraction of uPN input, grouped by type and lineage. The left group shows glomeruli that have only one uPN type, or one of the types for those with more than one. The right group shows the second uPN type for those glomeruli with more than one. Antennal lobe receptor neuron (ALRN) soma side indicated as 'ipsi' (ipsilateral), 'contra' (contralateral) or 'unk' (unknown, mostly corresponding to those glomeruli with fragmented ALRNs). Thermo/hygrosensory uPNs with suboesophageal zone (SEZ) innervation are indicated by 'Z' following the glomerulus. (**E**) Percentage of input onto uPN types relative to total connectivity (input + output), per lineage. Some of the outlier uPN types are labelled. Comparisons to categories with less than four data points were not done. Means per lineage were compared using Wilcoxon two-sample tests. Significance values: ns: p>0.05; *p≤0.05; **p≤0.01; ***p≤0.001; ****p≤0.0001. (**F**) Number of output synapses from olfactory or thermo/hygrosensory uPNs and mPNs. Number of neurons in each category shown at the top of the bar.

## Connectivity of olfactory projection neurons

Within the antennal lobe, ALPN dendrites connect with ALRN axons and ALLNs (*Figure 7A B*). As expected, olfactory mPNs and uPNs exhibit quite different connectivity profiles: mPNs receive both less overall dendritic input and also a smaller proportion of direct input from ALRNs than uPNs (30% vs. 50% comes from ALRNs). As a consequence of these connectivity profiles, uPNs show up earlier than mPNs in the layered olfactory system (*Figure 2E F*). In contrast, the connectivity profile of thermo/ hygrosensory ALPNs, of which 1/3 are biglomerular, is quite similar across ALPN classes and falls in between the olfactory uPNs and mPNs (*Figure 7C*).

When uPNs are broken down by type, we see a range of ALRN inputs (16–71%), the majority of them from ipsilateral ALRNs (for those with bilateral ALRNs) as well as from ALLNs (15–70%) (*Figure 7D*). In those glomeruli with more than one uPN type, the second uPN is usually from the GABAergic vPN lineage and receives significant input from the first, likely cholinergic uPN. vPNs (which include various multiglomerular PNs) provide feedforward inhibition to a range of targets in the LH (*Bates et al., 2020b*) and are thought to increase the fly's ability to discriminate (food) odours and gate between qualitatively different olfactory stimuli (*Liang et al., 2013*; *Parnas et al., 2013*). Curiously, the cholinergic V glomerulus uPN from the l2PN lineage (*Bates et al., 2020b*) resembles a vPN, both in terms of its output profile and total input fraction (*Figure 7D E*).

Although highly polarised, olfactory uPNs have hundreds of presynapses and thousands of outgoing connections from their dendrites while mPNs make far fewer connections. Thermo/hygrosensory ALPNs have very similar output profiles to each other, although thermo/hygrosensory mPNs, as with olfactory mPNs, provide much less output in the antennal lobe. The majority of these connections are onto ALLNs (56–75%), with the remaining being onto the dendrites of other ALPNs (*Figure 7F*).

## Higher-order olfactory neurons

The ALPN combinatorial odour code is read out by two downstream systems in very different ways. In general, the MB is necessary for the formation, consolidation and retrieval of olfactory memories, while other superior neuropils support innate olfactory processing (*Dubnau et al., 2001*; *Heimbeck et al., 2001*; *Krashes et al., 2007*; *McGuire et al., 2001*; *Parnas et al., 2013*; *Bates et al., 2020b*). This dichotomy is by no means absolute (*Dolan et al., 2018*; *Zhao et al., 2019*; *Yu et al., 2004*; *Séjourné et al., 2011*; *Sayin et al., 2019*; *Bräcker et al., 2013*), and indeed we find numerous examples of direct interactions between these brain areas (see also *Li et al., 2020*). Nevertheless, it remains a helpful simplification when investigating the logic innate vs. learned pathways.

Historically TOONs have often been defined by overlap with the axons of ALPNs. Using the hemibrain connectome we can now re-examine non-MB, TOON morphology exhaustively. We translated this into a connectomics definition of TOONs as 'neurons that receive either at least 1% (or 10 postsynapses in total) of their inputs from a single ALPN, or 10% of their inputs from any combination of ALPNs outside of the MB'. This revealed a total of ~2383 non-MB TOONs which means that both classic olfactory pathways – learned and innate – exhibit very similar convergence-divergence ratios: 2581:137:2035 ORN:PN:KC for the MB path and 2581:330:2383 ORN:PN:TOON for the non-MB path.

In the past, we focused on the LH when examining TOONs in the context of innate behaviour guidance (*Dolan et al., 2019*; *Frechter et al., 2019*), because the LH is the brain neuropil most heavily innervated by ALPNs (*Bates et al., 2020b*). Based on light-level data, we previously estimated ~1400 third-order LHNs forming >264 cell types (*Frechter et al., 2019*). The cell count estimate appears to have been accurate: of the hemibrain TOONs, ~ 60% (1428) have dendrites in the LH (*Figure 8A B*), making the LH the largest target for olfactory information beyond the antennal lobe (*Bates et al., 2020b*). With the higher resolution of the connectome, we were able to divide these LHNs into 496 near-isomorphic cell types (*Figure 8—figure supplement 1A*, see Materials and methods), many of which (~35%) could be matched to light-level data from the literature (*Frechter et al., 2019*). KCs, on the other hand, fall into only 15 types (*Li et al., 2020*). Therefore, in terms of cell types, the LH path exhibits far greater expansion than the MB path (*Caron et al., 2013*).

The distinction between LHNs and other TOONs remains useful in that it distinguishes a subset of TOONs that are part of the densely ALPN-innervated hub that is the LH. LHN cell types currently have more extant data in the literature, for example, allowing sparse genetic driver lines to be identified, or assignment of developmental identities and putative transmitter expression.

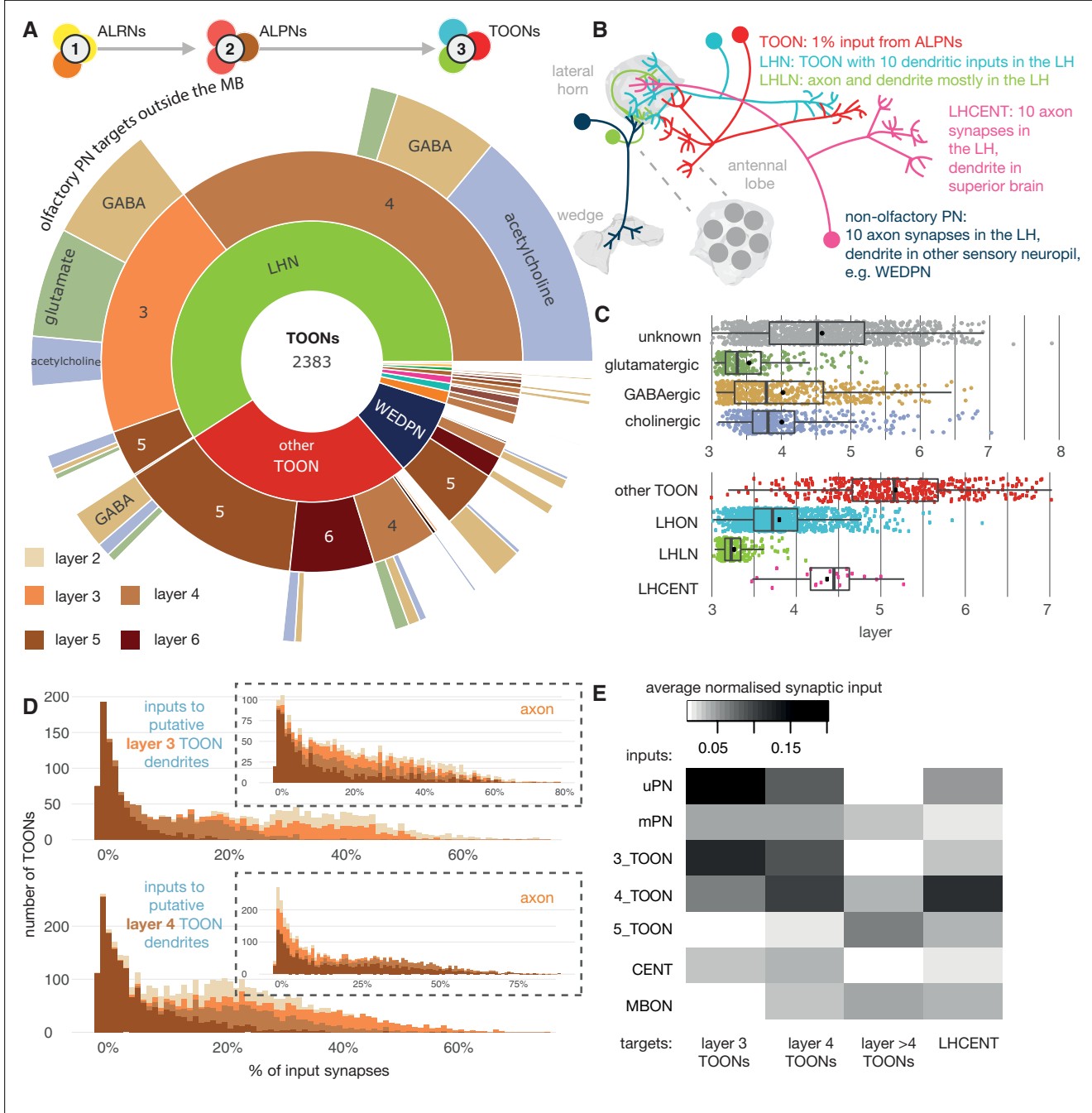

**Figure 8.** The targets of antennal lobe projection neuron (ALPN) axons. (**A**) Starburst chart breakdown of the 2383 targets of ALPN axons, outside of the mushroom body, by various properties. We term these neurons 'third-order olfactory neurons', or 'TOONs' (see text for definition). From the inside out, neurons are grouped by broad neuron class, layer according to the traversal model and their putative neurotransmitter. Most TOONs receive the majority of this input at their dendrites: green, lateral horn neurons (LHNs); dark blue, wedge projection neurons (WEDPNs); orange, dopaminergic neurons of the mushroom body (DANs); brown, descending neurons to the ventral nervous system (DNs); pink, lateral horn centrifugal neurons (LHCENTs). The starburst plot also includes some neurons connected only or mainly at their axons, including a small number of: light blue, visual projection neurons; yellow, severed contralateral axons; dark green, putative gustatory projection neurons from the gnathal ganglia; yellow, putative axons ascending from the ventral nervous system. (**B**) Schematic illustrating the definitions used to group neurons into broad classes. For details see Materials and methods. (**C**) Jitter plot showing olfactory layers of TOONs broken down by predicted transmitter (if known) and broad class (LHONs, LH output neuron; LHLN, LH local neuron) (*Frechter et al., 2019*). (**D**) The percentage of input supplied onto third-order neurons by different classes of input neuron. Upper: inputs onto third-order neurons' dendrite; lower: fourth-order neurons dendrites. Insets, input onto axons. (**E**) Normalised synaptic input to layer 3 and 4 neurons, as well as LH centrifugal neurons whose dendrites lie outside the LH but whose axons innervate it. Synaptic input is normalised by the total number of input synapses to the neuron's predicted axon or dendrite.

*Figure 8 continued on next page*

*Figure 8 continued*

The online version of this article includes the following figure supplement(s) for figure 8:

**Figure supplement 1.** Defining cell types for third-order olfactory neurons.

**Figure supplement 2.** Split-GAL4 lines for excitatory lateral horn output neurons.

**Figure supplement 3.** Split-GAL4 lines for inhibitory lateral horn output neurons.

**Figure supplement 4.** Split-GAL4 lines for lateral horn local neurons.

**Figure supplement 5.** Split-GAL4 lines for lateral horn input neurons.

With the benefit of a full, high-resolution LHN inventory from the hemibrain, we reassessed sparse genetic driver lines we previously generated to help experimentally target specific LHN cell types (*Figure 8—figure supplement 2*, *Figure 8—figure supplement 3*, *Figure 8—figure supplement 4*, *Figure 8—figure supplement 5*). We then grouped neurons into developmentally related 'hemilineages' and assigned all members of a given hemilineage the same 'transmitter identity' if we knew that at least one member of that hemilineage to express acetylcholine, GABA or glutamate based on immunohistochemical work (*Dolan et al., 2019*). Our assignments (*Figure 8C*, *Figure 8—figure supplement 1B*) are based on an assumption that neurons of a hemilineage share the same transmitter expression, as has been demonstrated for the ventral nervous system (*Lacin et al., 2019*). This is a useful proxy that gives an impression of fast-acting neurotransmitter expression diversity throughout the pool of TOONs, but it is far from definitive. We anticipate that machine learning methods will assist in automatic transmitter-type classification for synapses in data sets such as the hemibrain in the near future (*Eckstein et al., 2020*). LHNs are very diverse in terms of their hemilineage origins: ~30% of known hemilineages in the midbrain contribute to LHNs, with some more biased to layer 3 or layer 4 LHNs (*Figure 8D*). This is in contrast to KCs, which arise from a set of only four neuroblasts (*Truman and Bate, 1988*).

All the LHNs we consider are direct targets of olfactory ALPNs and would therefore historically be considered TOONs. In the absence of connectivity data, this is a necessary and useful simplification. Using the layers (*Figure 2C*), we can now for the first time take a more quantitative look at their putative position within the olfactory system. This shows that LHNs populate different layers of the olfactory system because the fraction of direct ALPN input can vary widely (*Figure 8C E*).

LHNs in layer 3 are mainly putative GABAergic or glutamatergic neurons based on their developmental origins and therefore likely inhibitory, while layer 4 LHNs are more commonly cholinergic and therefore excitatory (*Figure 8A*). It is important to note that the layer 4 LHNs are still direct synaptic partners of ALPNs; their designation as layer 4 is a result of weaker direct connectivity from ALPNs and slightly greater local input from layer 3 and 4 neurons (*Figure 8D*).

Matching hemibrain neurons to light-level data and partial tracings for neurons from FAFB shows that most 'anatomically' local neurons have a layer closer 3, and output neurons a layer closer to 4 (*Figure 8C*). The uPNs contribute most strongly and directly to the input budgets of layer 3 and 4 LHNs; in contrast, mPNs could be said to short-circuit the olfactory system, connecting to LHNs of layers 3–6 as well as other TOONs of the superior protocerebrum (*Figure 8* and Figure 10).

Individual TOON cell types can sample from a variety of ALPNs (*Figure 9*), and each type exhibits a relatively unique 'fingerprint' of input connectivity. Comparing the cosine similarity in ALPN→target connectivity between ALPN cell types reveals that uPNs and mPNs have very different connectivity profiles (*Figure 9—figure supplement 1*). While a certain amount of structure is present, there is no clear subgrouping of ALPN into subsets that serve as preferred inputs onto distinct target subsets. Thermo/hygrosensory ALPN cell types often exhibit similar connectivity with one another, and their uPNs clusters away from purely olfactory uPNs; however, their targets also commonly receive olfactory input from mPN cell types.

By breaking TOONs and their identified inputs into large classes (*Figure 8B E* and *Figure 10—figure supplement 1*), we can see that while direct uPN input to TOONs decreases from layers 3–5, mPN innervation remains constant and occurs onto both TOON dendrites and axons. Layer 3 TOONs heavily feedback onto ALPNs by making GABAergic axo-axonic contacts, while layer 4 TOONs feedback to layer 3 by both axo-dendritic and dendro-dendritic contacts.

If we think of obvious outputs of the olfactory system, we might consider dopaminergic neurons of the MB (DANs) or putative pre-motor DNs that project to the ventral nervous system, help to

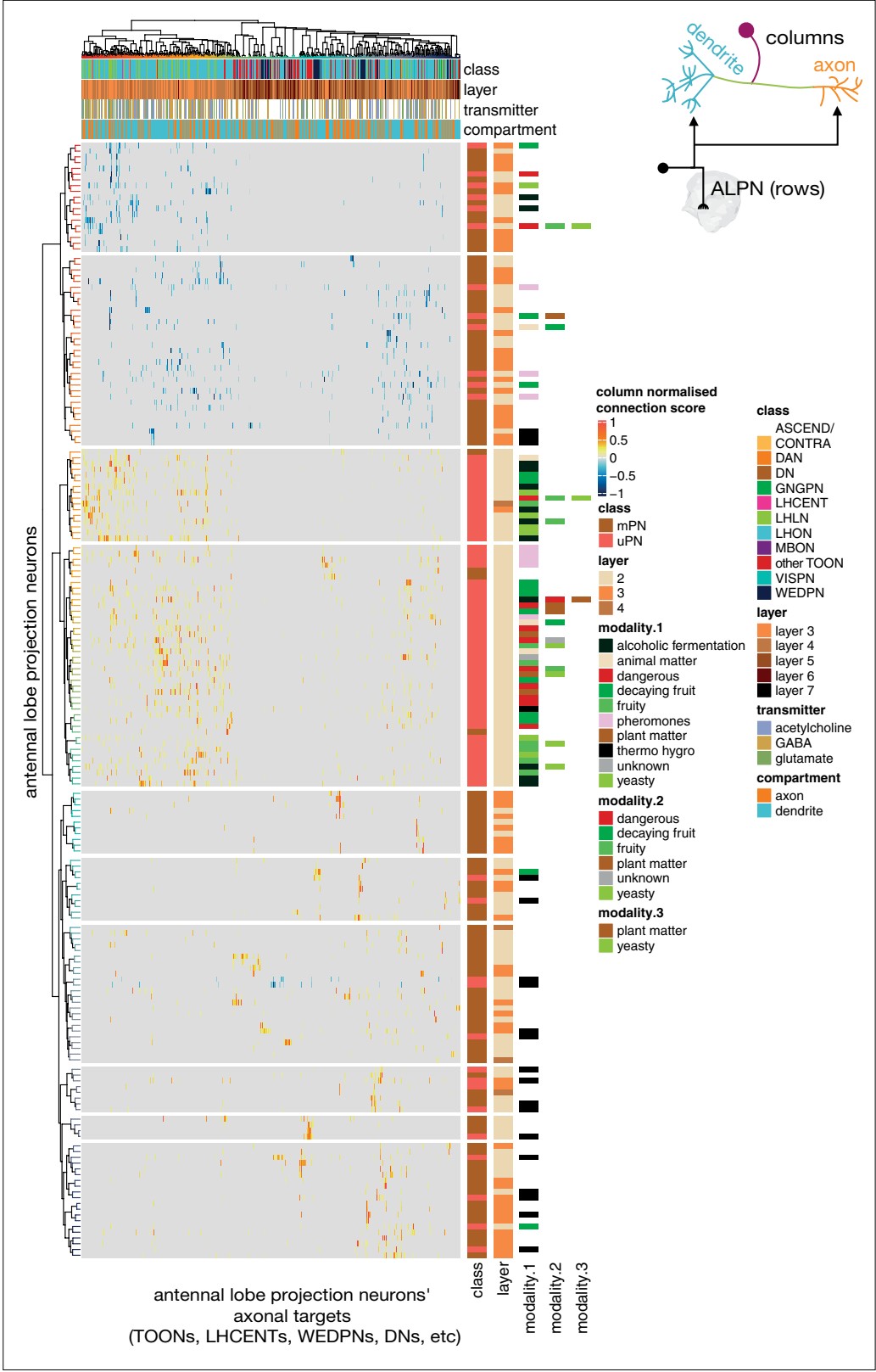

**Figure 9.** Antennal lobe projection neuron (ALPN) connectivity onto downstream targets. Annotated heatmap showing the ALPN cell types (188, rows) → target (column) connection strengths. These connection strengths have been max normalised per column (target). ALPNs known to be glutamatergic or GABAergic have been given negative connection strengths, those that are unknown or cholinergic, positive. Each target column represents

*Figure 9 continued on next page*

*Figure 9 continued*

an entire connectivity types' dendrites or axons (964 connectivity types' dendrites, 534 connectivity types' axons), in which each neuron has to have at least a 10 synapse or 1% postsynapse-normalised connection from an ALPN. Annotation bars indicate axons versus dendrites, as well as other metadata. Row and column clusters based on cosine similarity between connection strengths; see *Figure 9—figure supplement 1*. Where 'modality' is left white, the cell type in question combines information from multiple antennal lobe glomeruli. Clustering based on Ward's distance, ALPNs grouped into 10 blocks for visualisation.

The online version of this article includes the following figure supplement(s) for figure 9:

**Figure supplement 1.** Neurons at the antennal lobe projection neuron (ALPN) axon → target connection, clustered by connection similarity.

inform the writing of olfactory memory and the control of olfactory-related motor output, respectively. Strong output onto DANs and DNs first occurs with layer 4 TOONs and gets stronger with layer 5 TOONs, these contacts mostly being cholinergic axo-dendritic ones.

Higher TOON layers receive strong connections from memory-reading output neurons of the MB (MBONs) while lower ones receive greater, putatively inhibitory centrifugal feedback from neurons downstream of MBONs (LHCENTs) *Figure 8* and *Figure 10*. Using a neurotransmitter prediction pipeline based on applying machine learning to raw EM data of presynapses in the FAFB data set, LHCENT1-3, LHCENT5-6 and LHCENT9 appear to be GABAergic (*Eckstein et al., 2020*). LHCENT4 is predicted to be glutamatergic. LHCENT4 also differs from the others in that it is upstream of most other LHCENTs. LHCENT7 is predicted to be dopaminergic and has also been described as PPL202, a dopaminergic neuron that can sensitise KCs for associative learning (*Boto et al., 2019*).

## Stereotypy in superior brain olfactory neurons

Are these ~500 LHN types reproducible units? To address this question, we looked at the similarity in connectivity among members of the same cell type in the hemibrain data set (*Figure 11*). We also cross-compared hemibrain neurons with neurons in an EM volume of a different brain (FAFB) (*Figure 12*; *Zheng et al., 2018*). We find that 'sister' uPN – that is, those that have their dendrites in the same glomerulus and come from the same hemilineage – typically make similar numbers of connections onto common downstream targets. This is especially obvious when targets are grouped by their cell type rather than each considered as individual neurons (*Figure 11A–C*). Nevertheless, the consistency of these connections differ by sister uPN type, with some (e.g. DM4 vPNs, mean cosine similarity 0.50) being less similar to one another than a few non-sister comparisons (e.g. VC1 lPN and VM5v adPN, 0.63) (*Figure 11A*). For TOON cell types, comparing both up- and downstream connectivity to the axon or dendrite also yields a cosine similarity measure of ~0.75 (*Figure 11—figure supplement 1A*,B), with only a small difference between inputs/outputs and axon/dendrites (*Figure 11—figure supplement 1D*,E). The more similar the inputs to a cell type's dendrites, the more similar its axonic outputs (*Figure 11—figure supplement 1C*). Both also correlate with the morphological similarity between TOONs of a cell type (*Figure 11—figure supplement 1E*).

For comparisons with FAFB, we picked 10 larval-born 'secondary' hemilineages in the hemibrain data set and coarsely reconstructed all neurons of the same hemilineages in the FAFB volume (see Materials andmethods). We show that the morphologies can be matched between the two data sets and that, visually, these matches can be striking (*Figure 12*, *Figure 12—figure supplement 4A*). Every LHN and wedge projection neuron (*Bates et al., 2020b*) hemibrain cell type in these 10 hemilineages can be matched to one in FAFB (172 cell types), with some small variability in cell number per brain (*Figure 12*, *Figure 12—figure supplement 1*). We also examined a set of 'primary' embryonic-born neurons, the LH centrifugal neurons LHCENT1-11, and could match them up well between the two data sets. In some cases, putative cell types that appear isomorphic 'at light-level' can be broken down into several connectivity subtypes.

In several cases, we see that each of these subtypes have small but consistent morphological deviations between the two data sets (*Figure 12—figure supplement 2A*). To account for this, we broke our 569 morphological cell types into 642 connectivity types (*Scheffer et al., 2020*). In general, the closer the two neurons' morphology, the more similar their connectivity. However, similar morphologies can also have different connectivity (*Figure 12—figure supplement 4B*), perhaps due to non-uniform

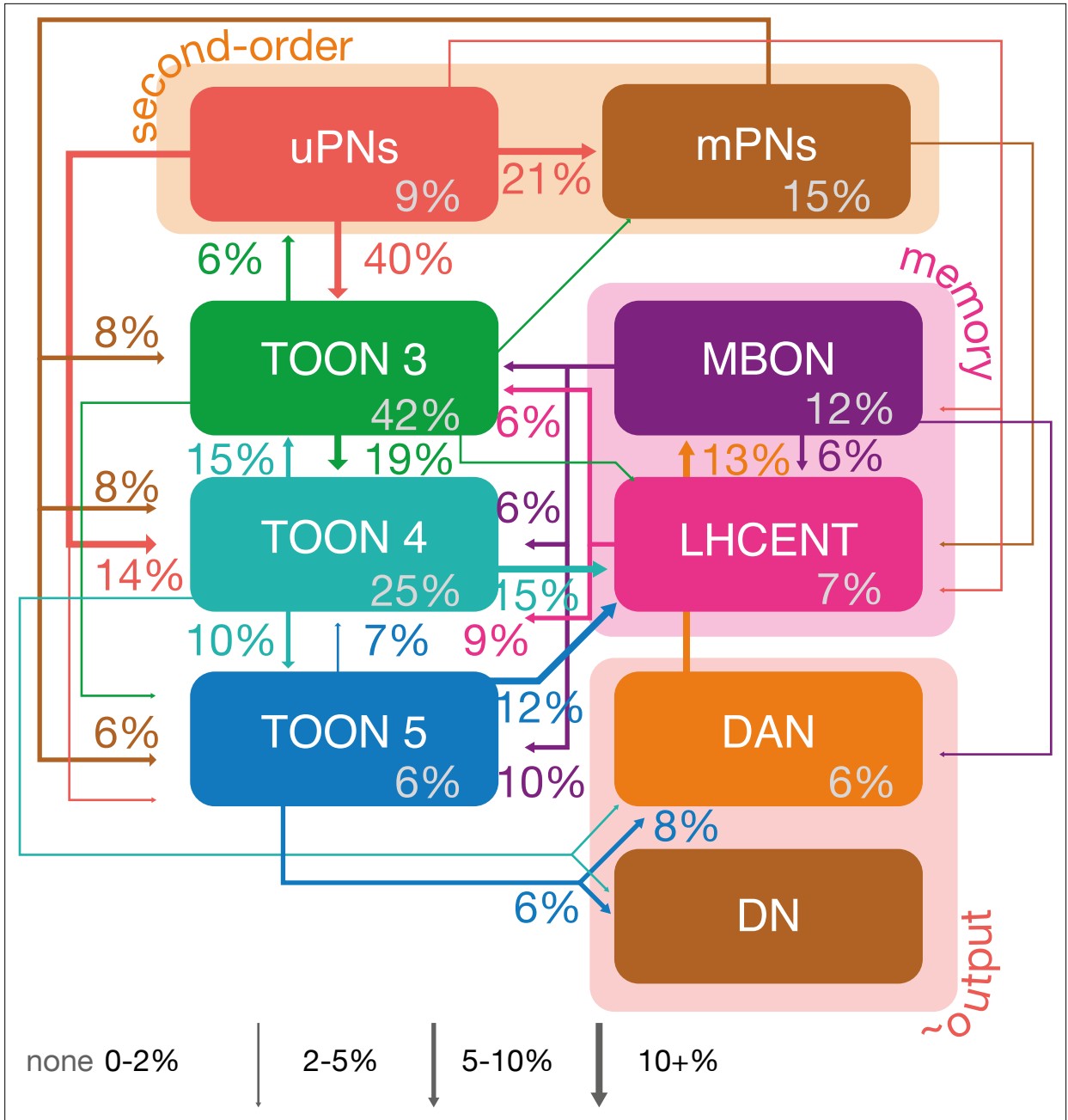

**Figure 10.** Neuron class-level network diagram of higher olfactory layers. A circuit schematic of third-order olfactory neurons, showing the average connection strength between different classes of neurons (mean percentage of input synapses), broken into their layers, as well as the antennal lobe projection neuron (ALPN), lateral horn centrifugal neuron (LHCENT) and mushroom body output neuron (MBON) inputs to this system and dopaminergic neuron (DAN) and descending neuron (DN) outputs. The percentage in grey, within coloured lozenges, indicates the mean input that class provides to its own members. The threshold for a connection to be reported here is 5%, and >2% for a line to be shown.

The online version of this article includes the following figure supplement(s) for figure 10:

**Figure supplement 1.** Neuron class-level network diagrams of higher olfactory layers, broken down by neuron compartments and putative transmitters.

under-recovery of synapses during the automatic segmentation of neurons and their connections in the hemibrain (*Scheffer et al., 2020*).

It is difficult to directly compare synapse numbers between the two data sets as the methods of reconstruction were very different (see Materials and methods). In FAFB, each human-annotated polyadic synapse has a mean of 11 postsynapses, whereas in the hemibrain machine annotation has resulted in ~8 (for the same, cross-matched neurons) (*Figure 12—figure supplement 4D*). This is

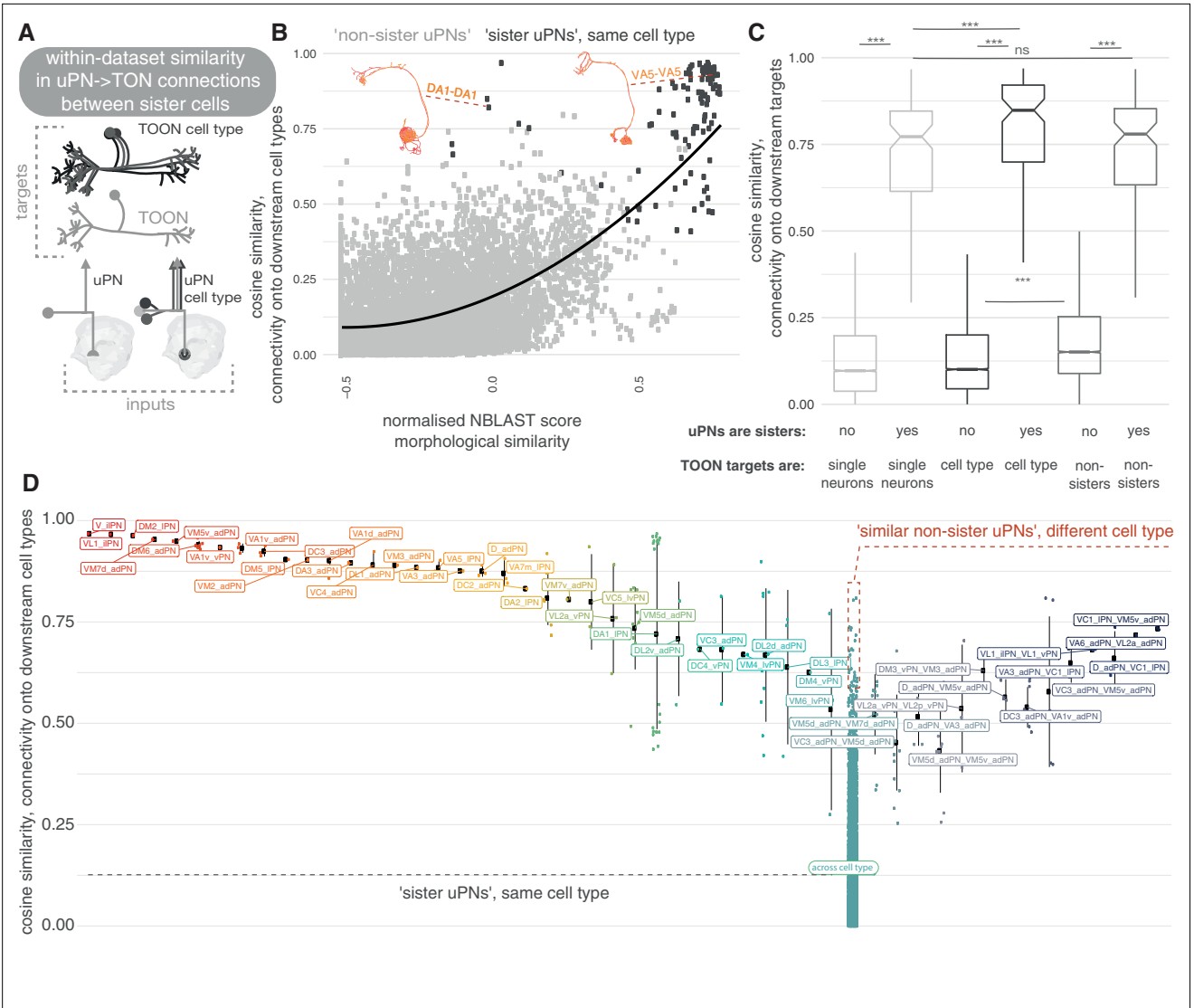

**Figure 11.** Within-data set connectivity similarity for key olfactory cell types. (**A**) The synaptic targets of uniglomerular projection neurons (uPNs) (left) and uPN cell types (right) can be thought of as both individual downstream cells (lower) as well as cell types (upper). (**B**) For each pair of uPNs, the cosine similarity for their outputs onto downstream cell types is compared against their morphological similarity. The uPN-uPN pairs where both neurons are from the same cell type, 'sisters', shown in dark grey, otherwise in light grey. (**C**) The cosine similarity in the downstream target pool for sister and non-sister uPN pairs is compared. Targets can either be considered as separate cells (light grey, leftmost boxplots) or pooled by cell type (dark grey, middle boxplots). Shuffled data, for which cell type labels were shuffled for neurons downstream of each uPN to produce random small out-of-cell-type groupings of cells, shown in mid grey (rightmost box plots). Non-sister third-order olfactory neurons (TOONs) are shuffled pairs of TOONs from different cell type. There are 113 different sister PN-PN comparisons and 9157 non-sister PN-PN comparisons from our pool of 136 uniglomerular PNs. (**D**) The cosine similarity between connections to downstream cell. Left, all reconstructed lateral horn neurons (LHNs) types, for uPN-uPN pairs. Pairs shown are from the same cell type (left) or different cell types, where at least one comparison has a similarity of above >0.6. Significance values, Wilcoxon test: ***p≤0.001.

The online version of this article includes the following figure supplement(s) for figure 11:

**Figure supplement 1.** Similarity in connectivity up- and downstream of olfactory neurons.

likely because different reconstruction methodologies have resulted in different biases for synaptic annotation. Nevertheless, we aimed to see whether ALPN→LHN connections in FAFB were also present in the hemibrain data set.

We previously reconstructed all members of selected cell types in FAFB (*Bates et al., 2020b*). Here, we manually reviewed the same types in the hemibrain data set (an average of three neurons per type) so that they are far more complete than the average hemibrain LHN (*Scheffer et al., 2020*)

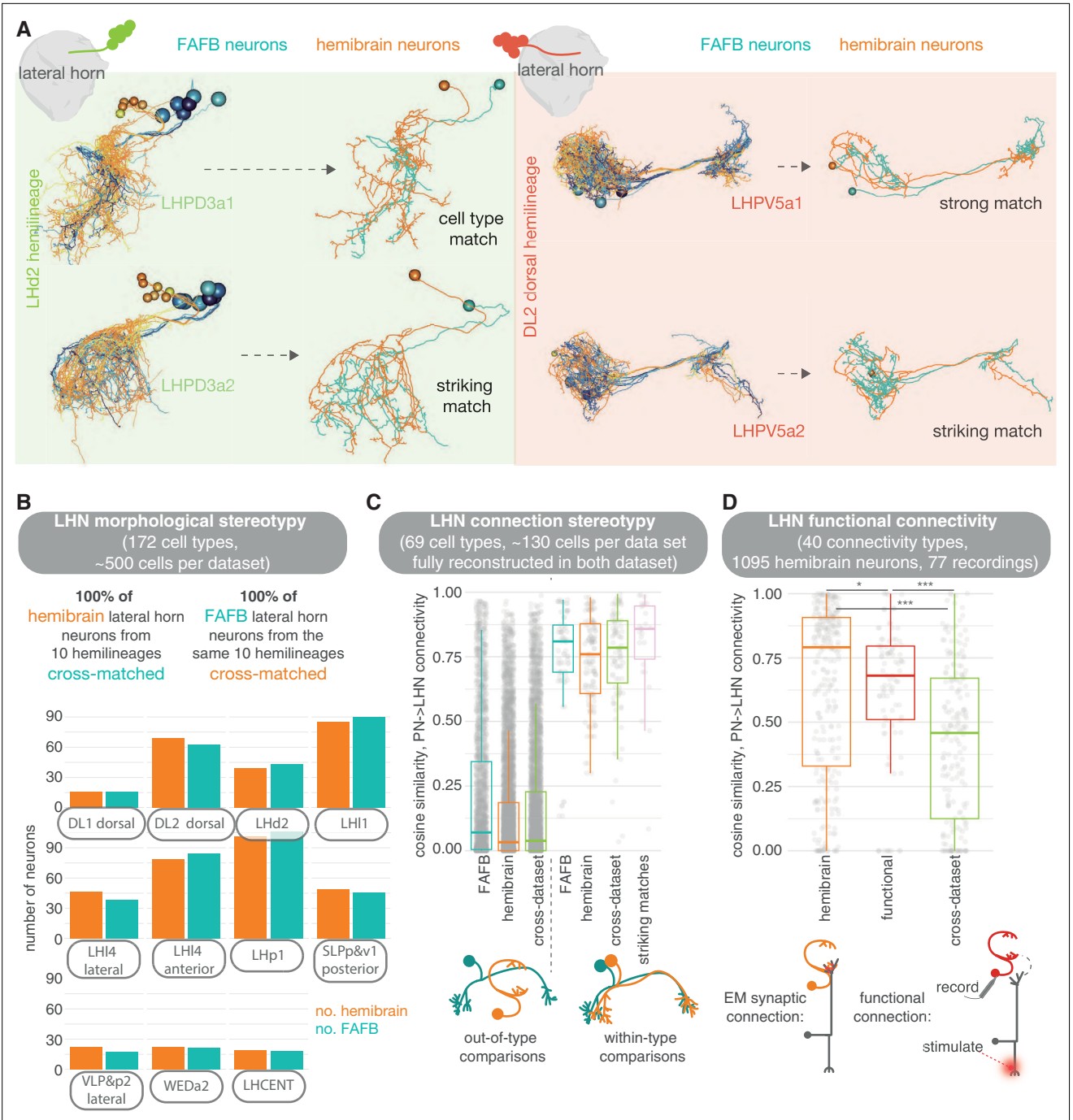

**Figure 12.** Stereotypy in morphology and connectivity between lateral horn neurons (LHNs) in the hemibrain, full adult fly brain (FAFB) and functional data sets. (**A**) Cell types and individual neurons that have been cross-matched between data sets. Examples from the hemilineages LHd2 (i.e. the dorsal most cell body group in the LHd2 lineage clone, otherwise known as DPLm2 dorsal) and DL2 dorsal (otherwise known as CP3 dorsal). (**B**) We were able to cross-match >600 neurons across 10 hemilineages between the hemibrain and FAFB. (**C**) For neurons that had been fully synaptically reconstructed in FAFB, we calculate the cosine similarity for their antennal lobe projection neuron (ALPN) → LHN connectivity vectors to hemibrain neurons, both out-of-cell-type (left) and within-cell-type (right), as well as between the two data sets. In pink, same-cell-type between data set comparisons are made for only our 'best' morphological matches; matches for which the two neurons look so similar they could be the 'same cell'. (**D**) Within-cell-type cosine similarity for ALPN→LHN connectivity for within the hemibrain data set, within the *Jeanne and Wilson, 2018* functional connectivity data set and between members of the same cell type across data sets. Significance values, Student's t-test: ns: p>0.05; *p≤0.05; **p≤0.01; ***p≤0.001; ****p≤0.0001.

The online version of this article includes the following figure supplement(s) for figure 12:

**Figure supplement 1.** Stereotypy in morphology between lateral horn neurons (LHNs) in the hemibrain and full adult fly brain (FAFB) data sets.

*Figure 12 continued on next page*

Figure 12 continued

**Figure supplement 2.** Stereotypy in connectivity between lateral horn neurons (LHNs) in the hemibrain and full adult fly brain (FAFB).

**Figure supplement 3.** Stereotypy in connectivity between lateral horn neurons (LHNs) in the hemibrain and a functional data set.

**Figure supplement 4.** Matching synaptically complete neurons between two EM data sets.

(see Materials and methods). We also examined other cell types for which we have only subsets in FAFB (*Figure 12—figure supplement 4A*). Normalised connections strengths (normalised by total input synapses) from ALPNs to LHNs are, on average, stronger in the hemibrain than in FAFB. In the hemibrain, a larger total number of input synapses have been assigned per neuron but fewer ALPN→LHN connections, perhaps an artefact of the different reconstruction methods employed (*Figure 12—figure supplement 4C*). Nevertheless, by comparing our FAFB reconstructions with their cognates in the hemibrain for 12 connectivity types, using a cosine measure for connection similarity, we see that the variability in ALPN→LHN connections between data sets is no greater than within the same data set (*Figure 12*, *Figure 12—figure supplement 2B*).

This suggests that morphological cell types may be as consistent between animals as within an animal. We also compare the hemibrain connectivity to a data set describing functional connectivity between antennal lobe glomeruli and LHNs (*Jeanne and Wilson, 2018*). For some LHNs, these functional connections are well recapitulated in the hemibrain's cognate uPN→LHN synaptic connectivity. For many other pairs, however, the connectivity similarity is no greater than that to other neurons in the data set (*Figure 12*, *Figure 12—figure supplement 3*): some functional connections are not present as direct synaptic connections in the connectome and vice versa. Similarly, there is no clear correlation between the strength of a functional connection and the synaptic strength of corresponding hemibrain ALPN→LHN connections (*Figure 12—figure supplement 3D,E*). This could be due to the action of local processing in the LH as well as connections from mPNs, which have impacted feedforward transmission more for some LHN cell types than for others. For example, LHAV4a4 neurons have very similar structural and functional connectivity, while LHAV6a1 neurons do not, though both their structural and functional connectivity seem stereotyped even if they are different from one another (*Jeanne and Wilson, 2018*; *Fişek and Wilson, 2014*). In addition, functional connection strength integrates inhibitory and excitatory inputs from different ALPN classes, which might also confound our results. Indeed, the glomeruli for which we have some of the largest deviations from the hemibrain structural data are those with GABAergic uPNs (*Figure 12—figure supplement 3B*).

## Integration of innate and learned olfactory pathways

With the hemibrain data set, we can look at the extent to which MBONs directly connect to LHNs. We see that while most olfactory ALPN input is onto LHN dendrites, most MBON input is onto their axons (*Figure 13*). We quantify this using an ALPN-MBON axon-dendrite compartment separation score (see Materials and methods) and find high compartmental segregation of inputs, with MBONs inputting onto LHN axons (though many cells have a score at or near zero as they receive little MBON innervation) (*Figure 13—figure supplement 4*). Many of those with negative scores are either neurons tangential to the LH or LH centrifugal neurons, whose MBON innervation is known to target their dendrites (*Bates et al., 2020b*). More than 20% of layer 4 LHN axons are targeted by a range of MBONs (*Figure 13*): both cholinergic and GABAergic, and including MBONs implicated in both aversive and appetitive learning (*Aso et al., 2014b*). MBON connectivity to LHNs is sparse and only a few LHNs receive inputs from multiple MBONs (*Figure 13*). With MBON→LHN connections being axo-axonic, there is the potential of them being reciprocal. However, there is very little output from LHNs onto MBON axons (*Figure 13*), suggesting that MBONs might gate LHN activity, but not vice versa.

Next, we asked whether MBONs target the axons of LHNs that pool particular kinds of olfactory information. To examine this question, we performed a matrix multiplication between connectivity matrices for ALPN→LHN dendrite innervation, and MBON→LHN axon innervation, normalised by the LHN compartment's input synapse count, to generate a 'co-connectivity' score (*Figure 13—figure supplement 1C,D*). From this, three coarse groups emerge: some MBON types seem to preferentially target 'putative food-related' LHNs. These LHNs receive input from ALPNs that respond to mostly yeasty, fruity, plant matter and alcoholic fermentation-related odours. Another group preferably targets a separate set of LHNs, which themselves receive input from ALPNs involved in thermosensation,

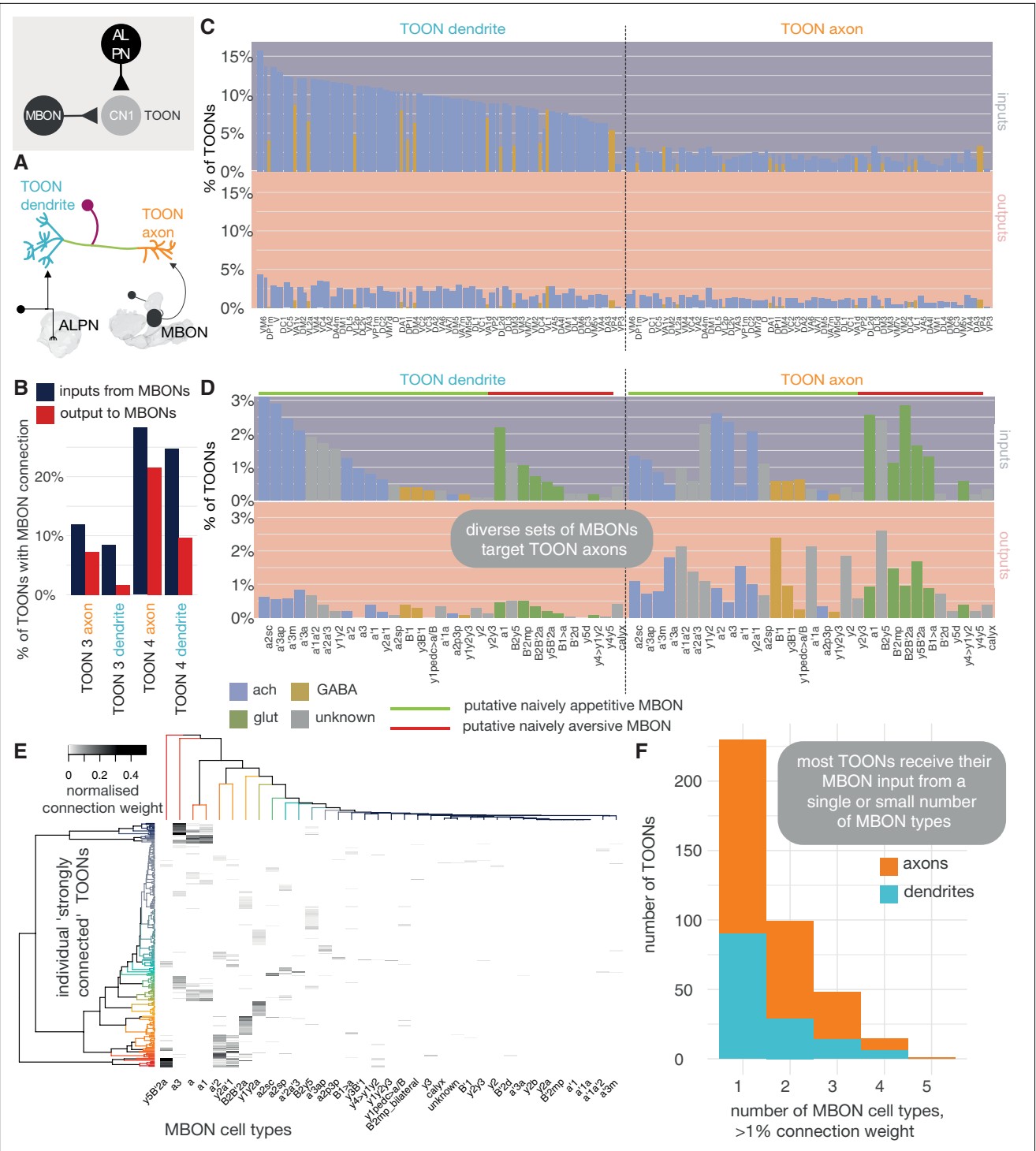

**Figure 13.** Mushroom body output neuron (MBON) innervation of lateral horn neurons. (**A**) Olfactory projection neurons and MBONs seem to target different ends of lateral horn output neurons. (**B**) The percentage of third-order olfactory neurons (TOONs) (2383 neurons in total) that receive a 'strong' connection from an MBON type (71 neurons in total) (>1% of their dendrite's/axon's input synapses). (**C**) Percentages are broken down by MBON cell type. (**D**) The percentage of TOONs that receive a 'strong' connection from a uniglomerular projection neuron (uPN) type (136 neurons in total), broken down by type (>1% of their dendrite's/axon's input synapses). (**E**) A heatmap showing the normalised input of different MBONs onto TOONs' axons. (**F**) A histogram showing the number of downstream TOONs that receive input from different numbers of MBONs. A threshold of >1% the input synapse count is used, axons and dendrites treated separately.

The online version of this article includes the following figure supplement(s) for figure 13:

*Figure 13 continued on next page*

*Figure 13 continued*

**Figure supplement 1.** Propagating known odour information to third-order olfactory neurons (TOONs) and mushroom body output neurons (MBONs).

**Figure supplement 2.** An exemplar convergence cell type of the lateral horn (LH) and mushroom body (MB).

**Figure supplement 3.** Convergence neurons of the lateral horn (LH) and mushroom body (MB).

**Figure supplement 4.** A class-compartment separation score.

ethanol, $CO_2$, aversive fruity odours and pheromones. The third pool of MBONs wire with neurons from both pools of ALPNs. About half the uPNs did not have a strong co-connectivity score with MBONs. To try and assess whether certain MBONs might play a role in the processing of particular odours, we multiplied the co-connectivity matrix by odour response data from a recent study (*Badel et al., 2016*). We did not see a striking separation, though all MBONs converge on TOONs that get appetitive fruity odours (e.g. ethyl butyrate) information from PNs, largely because these odours are well represented on the PN level, and less so highly specific odours that are less broadly encoded (*Figure 13—figure supplement 1A*), such as the bacterial odour geosmin.

In examining neurons downstream of MBONs, we found a cell type of 12 neurons which receives an unusually high proportion, up to ~37%, of their input connections from MBONs: LHAD1b2, cholinergic LH output neurons whose activation generates approach behaviour (*Dolan et al., 2019*; *Frechter et al., 2019*). Electrophysiological recording of these cells has shown them to act as a categoriser for 'rotting', amine-type odours (*Frechter et al., 2019*). Consistent with connectivity observed in FAFB (*Bates et al., 2020b*), we find now the full suite of excitatory, naively aversive and inhibitory appetitive MBONs that target LHAD1b2 axons, and the naively appetitive MBONs and specific ALPNs that target their dendrites (*Figure 13—figure supplement 2A,B*). We also observe LHAD1b2 connections onto the dendrites of PAM DANs involved in appetitive learning, again consistent with work in FAFB (*Otto et al., 2020*; *Figure 13—figure supplement 2C*). Together, this builds a model whereby naively appetitive information from the LH signals the presence of rotting fruit (*Mansourian and Stensmyr, 2015*). This activity is then bidirectionally gated by MBON input: expression of an aversive memory reduces the cholinergic drive to the axon, while an appetitive memory reduces glutamatergic inhibition, thereby potentiating the cell type's effect on its downstream targets. If the cell type fires, it could excite PAM DANs that feedback to create a long-term depression in MB compartments associated with naive aversion, that is, appetitive learning.

The next level at which 'innate' information from the non-MB arm of the olfactory system and 'learned' information from the MB arm can converge is in 'convergence' neurons (CN2) downstream of both of these neuropils. By looking at LHN cell types known to evoke either aversive or appetitive behaviour (*Figure 13—figure supplement 3A*; *Dolan et al., 2019*), we see that downstream partners of appetitive LHNs are more likely to be innervated by MBONs than those of aversive LHNs (*Figure 13—figure supplement 3C*). CN2 neurons that receive at least 1% of their synaptic inputs from LHNs or from MBONs tend to get cholinergic input from naively appetitive MBONs and LHNs, and inhibitory input from naively aversive MBONs and LHNs (*Figure 13—figure supplement 3B,D*).

## Connections to the motor system

Motor systems ultimately responsible for generating behaviour are located in the ventral nervous system and the suboesophageal zone (SEZ) and can, to some extent, function independently of the rest of brain (*Berni et al., 2012*; *Hückesfeld et al., 2015*; *Egeth, 2011*; *Hampel et al., 2017*). How olfactory circuits connect to and modulate these motor systems remains an open question. In general, higher brain circuits exert control over motor systems via DNs (*Lemon, 2008*). In *Drosophila*, a recent light-level study identified ~700 DNs (~350 per side of the brain) that connect the brain to the ventral nervous system (*Namiki et al., 2018*). We used existing neuPrint annotations and complemented them with DNs identified in the 'FlyWire' segmentation of FAFB to compile a list of 345 confirmed DNs in the hemibrain data set (see Supplementary files; *Dorkenwald et al., 2020*). Due to the truncation, the hemibrain volume does not contain many of the DNs in the SEZ ('DNg' in *Namiki et al., 2018*) and most of the DNs present descend from higher brain regions. Even without knowing their exact targets in the ventral nervous system, such DNs represent a common outlet for all higher brain circuits. We find only 11 DNs that appear to be 'early' (i.e. layer 3 or 4) with respect to the olfactory system (*Figure 14*). These early DNs typically receive diverse inputs including from ALPNs and LHNs

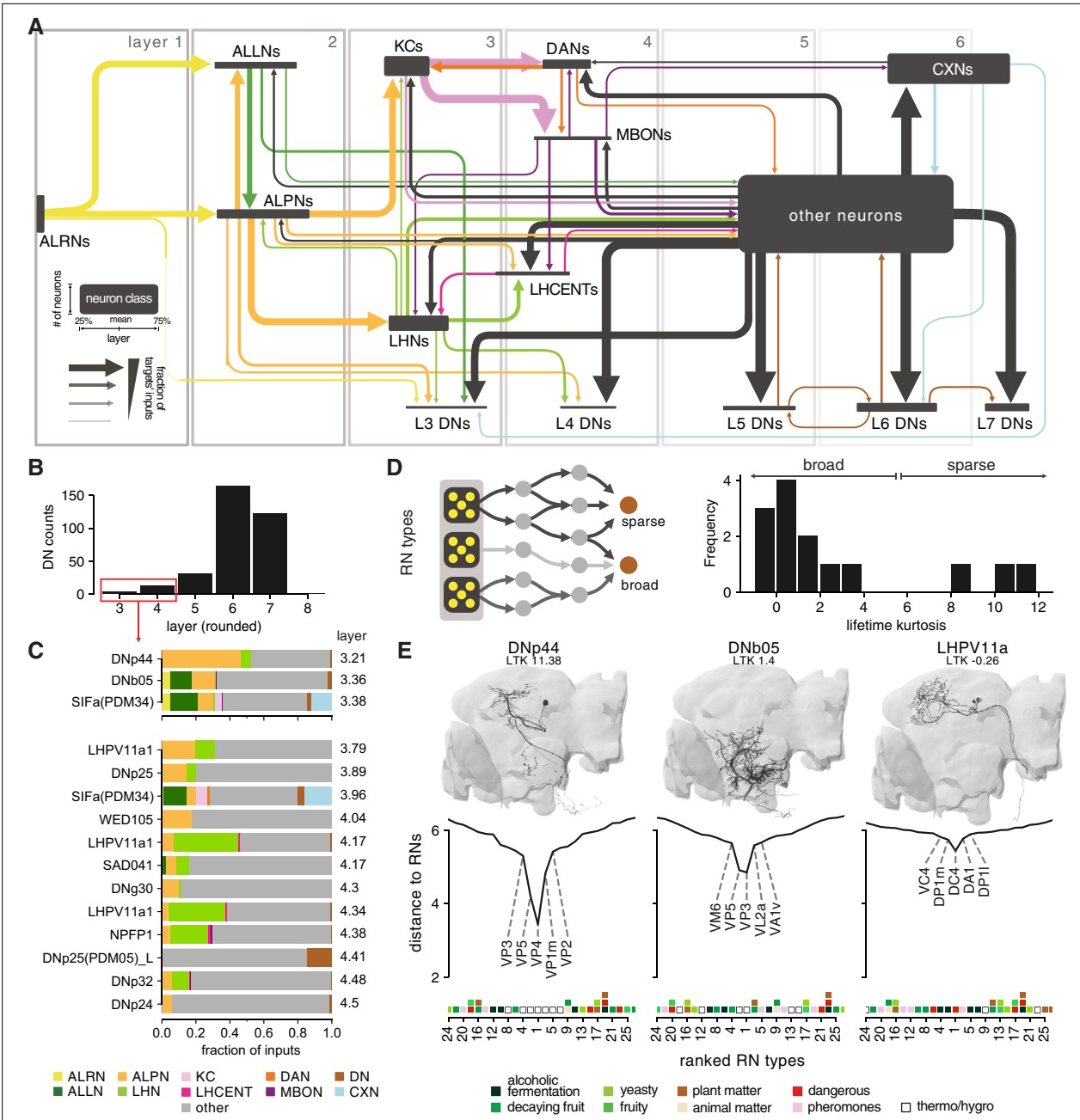

**Figure 14.** Connections between the olfactory system and descending neurons (DNs). (**A**) Summary of olfactory circuits organised by layers. Box heights and widths correspond to the number and layer of neurons represented, respectively; arrow widths correspond to fraction of the targets' inputs. See also legend in lower left. (**B**) The number of 'early' (layers 3 and 4) DNs is low. (**C**) Inputs to early DNs are diverse. Labels represent names in neuPrint. (**D**) Sparseness (lifetime kurtosis, LTK) of early DNs with respect to individual receptor neuron (antennal lobe receptor neuron [ALRN]) types. Most early DNs receive indirect inputs from a broad range of ALRNs. (**E**) Exemplary DNs and their connectivity to individual ALRN types. A low distance indicates a more direct connection between an olfactory receptor neurons (ORNs) or thermo-receptor neurons (TRN)/hygro-receptor neurons (HRN) type and the DN. Only the top 25 ALRN types shown. Hemibrain DNs are shown in black, and their homologs in the FlyWire data set as reference in grey. Heatmap shows glomeruli odour scenes.

The online version of this article includes the following figure supplement(s) for figure 14:

**Figure supplement 1.** Extended data for *Figure 14*.

(*Figure 14*). We next asked whether individual DNs exhibit preferences with respect to which types of ALRNs they receive direct or indirect input from. To answer this, we re-ran the graph traversal model using only the ALRNs of a given type/glomerulus as seeds. This produced, for each DN, a

vector describing the distances to 49 different ALRN types (we excluded some of the more severely truncated glomeruli). Using those vectors to calculate the lifetime kurtosis, we find both broad and sparse early DNs (*Figure 14*). By contrast, DNs in layer 5 and above are generally broadly tuned and no longer exhibit a preference for specific ALRNs (data not shown). There do appear to be 'shortcuts' between the thermo/hygrosensory and the motor system via early DNs that connect most directly to VP1-5 ALRNs. One might expect similar connections for biologically highly relevant odours such as the wasp pheromone Iridomyrmecin (Or49a/Or85f, DL4), Geosmin (Or56a, DA2) or the sex pherhomone cVA (Or67d, DA1) (*Mansourian and Stensmyr, 2015*; *Stensmyr et al., 2012*; *Kurtovic et al., 2007*; *Kohl et al., 2013*; *Ebrahim et al., 2015*). However, ORNs appear to only converge onto broadly tuned early DNs that show no clear preferences for specific odour scenes (*Figure 14* and *Figure 14—figure supplement 1*). This suggests that thermo/hygrosensation employs labelled-line shortcuts, whereas olfaction uses (higher-order) population coding to effect motor output.

## Discussion

One of the most significant practical outcomes of our work are classifications for thousands of olfactory system neurons across the hemibrain data set, comprising a full inventory for a single brain hemisphere (see Supplemental material). This includes the first full survey of ALLNs, TOONs and LHCENTs, and complements a recent inventory of ALPNs (*Bates et al., 2020b*; *Figure 1*). We explore this data with a model that breaks down the olfactory system into layers. Layering had not previously been computable for higher-order neurons, and this analysis reveals interesting features even within the first three layers. Additionally, we have investigated high-level connectivity motifs between the neuron classes and cell types that we have defined and examined how stable our classifications are by asking whether we can find the same neurons, and in some cases the same connections, in a second connectomic data set.

### Cell-type annotations across the first three orders of the olfactory system

We have built open-source neuroinformatic tools in R and Python (see Materials and methods) to read and summarise neuron data from the hemibrain data set efficiently. We have used these with morphological clustering tools, namely NBLAST (*Costa et al., 2016*), to break neurons into groups that we can validate against other neuron data, both from light microscopy (*Chiang et al., 2011*) and another EM data set (*Zheng et al., 2018*). In so doing, for the right hemisphere, we have classified all 2644 receptor neurons (ALRN, olfactory and thermo/hygrosensory) in all 58 antennal lobe glomeruli, as well as the 338 second-order projection neurons (uPNs and mPNs) and 196 ALLNs, and 2300 third-order neurons outside of the MB. We connect these olfactory neurons to known cell types, and for ALLNs (*Figure 6E*) and LHNs we have expanded extant naming systems to cover hundreds of new morphologies (*Figure 8A*). For the whole hemibrain data set of ~25,000 neurons, we assign a putative olfactory layer (*Figure 2*). We find that for layers 1–3, information is mostly propagated forward, for layers 4–6 there is much intra-layer cross-talk, and from 7 onwards information tends to propagate back to lower layers (*Figure 2G*). In light of this new data, we have also re-evaluated the neurons targeted by recently published LH split-GAL4 lines (*Dolan et al., 2019*; *Figure 8—figure supplement 2*, *Figure 8—figure supplement 3*, *Figure 8—figure supplement 4*, *Figure 8—figure supplement 5*).

### Class-level connection motifs in the olfactory system

We have found that connectivity with respect to first-order olfactory inputs, the ALRNs, differs depending on whether the axon enters the antennal lobe from the ipsi- or contralateral side of the brain (*Figure 3*). Although there have been functional indications of asymmetric information processing (*Gaudry et al., 2013*), no connectomic signature had been observed in adult *Drosophila* before, while in larva ORNs are unilateral. We identify a general principle that ipsilateral sensory input has stronger feedforward connections to the ALPNs that convey information to higher centres, while contralateral ALRNs are biased to form connections with ALLNs. We also show specific connectivity motifs such as the extreme bias for contralateral sensory input of the broadly innervating bilateral il3LN6 neurons, which appear to be the adult analogue of the larval 'Keystone' (*Berck et al., 2016*) ALLNs

(*Figure 5B E*). We see that many sparse ALLNs innervating a small number of glomeruli interact specifically with thermo/hygrosensory circuits; although this is consistent with a model in which these seven glomeruli form a specialised subsystem, there are local interactions with other glomeruli so they are not completely isolated. Furthermore, some ALLN cell types are segregated into axon and dendrite, which facilitates reciprocal interactions between, for example, the 'heating' glomerulus VP2 and the 'cooling' glomerulus VP3. The antennal lobe also receives feedback from superior brain regions, and this primarily targets the ALLN network, as opposed to ALPN dendrites or ORN axons (*Figure 5G*).

Amongst ALPNs, we see a second general rule: while uPNs mostly receive feedforward input, multiglomerular mPNs get a higher proportion of their input from lateral ALPN-ALPN connections and from ALLNs, meaning that in our analysis many emerge as layer 3 neurons (*Figure 2E F*). The uPNs provide most of the feedforward drive to the TOONs. However, they provide decreasing levels of input to TOONs from layer 3 to layer 5. They receive feedback to their axons from largely glutamatergic or GABAergic layer 3 TOONs (cells we once classed as LH local neurons) and LH centrifugal neurons. We expect these connections to inhibit uPN axons. The mPNs can short-circuit this progression and provide roughly consistent amounts of input to all groups of TOONs, both at their dendrite and axons. Comparison with our recent work reveals that we had previously thought of layer 3 TOONs as 'local' neurons and layer 4+ LHNs as 'output' neurons (*Figure 8E*). As olfactory information filters through to layer 5+ TOONs, stronger connections are made to 'outputs' of the olfactory system, including dopaminergic neurons that can inform memory and DNs that contact premotor circuits (*Figure 10—figure supplement 1*).

These output neurons can get strong but sparse input from a diversity of MBONs to their axons, acting as 'convergence level 1' (CN1) neurons that re-connect the non-MB and MB arms of the olfactory system (*Figure 13*). This MBON innervation is biased towards TOONs that receive input from certain ALPN groups, including those that encode food-like odours (*Figure 13*). Neurons downstream of TOONs can also receive MBON input; these are 'convergence level 2' (CN2) neurons. There are more CN2 neurons downstream of known appetitive TOONs than aversive ones (*Figure 13—figure supplement 1F*). In general, CN2 neurons tend to get inhibitory inputs from naively aversive MBONs and TOONs, and excitatory input from naively appetitive MBONs and TOONs (*Figure 13—figure supplement 3*). Analogous innate-learned integration has been studied in the larva, also in connectome-informed experimentation (*Eschbach et al., 2020*). The authors investigated a CN2 cell type and found it to be excited by appetitive LHNs and MBONs and inhibited by aversive MBONs. Naive MBON activity is likely to be relatively stereotyped between animals (*Mittal et al., 2020*). The hypothesis is that in naive animals opposing MBON drive balances to produce a stereotyped 'innate' outcome; learning then shifts this balance to bias behaviour.

## Between-animal stereotypy in olfactory system neurons

One of the most pressing questions for the field now is how stereotyped the fly brain actually is. This is critical for interpreting connectomes, but also a fundamental issue of biology across species all the way to mammals. We do not expect two fly connectomes to be exactly the same. However, there is a palpable expectation that one would identify the same strong partners for a neuron of experimental interest or reveal a shared architecture of some circuit because many small cell types are faithfully reproduced between animals (*Bates et al., 2019*).

Here, we have found that all ALPN cell types from a complete survey in FAFB could be found in the hemibrain, with small variations in cell number that correlate with birth order (*Figure 6C* and *Figure 6—figure supplement 1A*). More variation occurs in the number of larval-born secondary neurons than the primary neurons born in the embryo. There are several possible reasons for these differences, including the fact that in the larva each of the 21 olfactory glomeruli is defined by a single ORN and ALPN. Since missing one neuron would therefore eliminate a whole olfactory channel, there might be a strong drive to ensure numerical consistency.

Assessing cell-type stereotypy of mPNs and ALLNs between hemibrain and FAFB is somewhat compromised by truncation of glomeruli in the hemibrain data set. However, examining morphologically far more diverse LHNs, we could find the same cell types across 10 hemilineages in similar numbers (*Figure 12—figure supplement 1*).

Because LHNs also have reasonably stereotyped dendritic projections (*Dolan et al., 2019*), functional connections from ALPNs (*Jeanne and Wilson, 2018*) and responses to odorants (*Frechter*

*et al., 2019*), it is likely that ALPN-LHN contacts have intrinsic relevance to the animal. Conversely, olfactory ALPN-KC contacts have minimal intrinsic meaning and exhibit near-random wiring (*Eichler et al., 2017*; *Zheng et al., 2020*; *Caron et al., 2013*), although connection biases may enable associative memory to focus on certain parts of olfactory space (*Zheng et al., 2020*). ALPN connectivity onto third-order neurons in the 'non-MB' path through the olfactory system appears to be reasonably stereotyped, as suggested by the strong morphological stereotypy among these higher-order neurons (*Figure 12*). Structural connectivity from the hemibrain does not necessarily capture functional connections assayed by physiology. Encouragingly, however, recent work with a retrograde genetic system for finding neurons that input onto genetically targetable cells found 6/7 glomerular connections to LHPD2a1/b1 neurons of above 10 synapses in FAFB, and 8/9 for LHAV1a1 (*Cachero et al., 2020*).

## Conclusion

Our study (together with the work of *Li et al., 2020* on the MB) provides an annotated guide to the complete olfactory system of the adult fly. We believe that it will be invaluable in driving future work in this important model system for development, information processing and behaviour. Our microcircuit analysis already raised specific hypotheses about brain functions including stereo processing of odours, higher-order feedback controlling sensory processing and the logic of integration downstream of the two main higher olfactory centres.

The tools and analytic strategies that we have developed should enable many future analyses of the hemibrain data set as well as in progress and planned data sets for the male and female central nervous system. For example, the layer analysis could usefully be carried out across sensory modalities to quantify multisensory integration. They also provide a quantitative basis for comparative connectomics studies across data sets, for which we provide initial comparisons at two different levels of

**Table 1.** R and Python packages used and developed in this study.

|  | Language | Name | Github repository | Versioned DOI | Description |
|---|---|---|---|---|---|
| By the authors | R | neuprintr | natverse/neuprintr | 10.5281/ zenodo.3843545 | Query data from neuPrint |
|  | R | hemibrainr | natverse/hemibrainr | 10.5281/ zenodo.4969908 | Analyse hemibrain data and metadata |
|  | R | catmaid | natverse/rcatmaid | 10.5281/ zenodo.3357143 | Query CATMAID data (e.g. for FAFB) |
|  | R | nat.jrcbrains | natverse/nat.jrcbrains | 10.5281/ zenodo.4966564 | Map between brain templates (inc hemibrain and FAFB) |
|  | R | nat.nblast | natverse/nat.nblast | 10.5281/ zenodo.4966689 | Morphological comparison |
|  | Python | navis | schlegelp/navis | 10.5281/ zenodo.4751181 | Query and process neuron data |
|  | Python | navis-flybrains | schlegelp/navis-flybrains | 10.5281/ zenodo.4966641 | Map between brain templates (inc hemibrain and FAFB) |
|  | Python | pymaid | schlegelp/pymaid | 10.5281/ zenodo.4724559 | Query CATMAID data (e.g. for FAFB) |
|  | Python | fafbseg | flyconnectome/fafbseg-py | 10.5281/ zenodo.4966610 | Work with autosegmented FAFB data (e.g. FlyWire) |
| Third party | Python | neuprint-python | connectome-neuprint/ neuprint-python | 10.5281/ zenodo.4968061 | Query data from neuPrint, developed by Stuart Berg (Janelia Research Campus) |

the olfactory system. Finally, these strategies and the circuit principles that they uncover provide a platform for connectomics approaches to larger brains that will surely follow (*Abbott et al., 2020*).

## Materials and methods

### Data and tool availability

Hemibrain versions 1.1 and 1.2 data are available via neuPrint (https://neuprint.janelia.org/) (*Clements et al., 2020*; *Scheffer et al., 2020*). New FAFB tracing data presented in this study will be made available through the public CATMAID instance hosted by Virtual Fly Brain (https://fafb.catmaid.virtualflybrain.org/) upon publication. Previously published FAFB data is already available on the site. FlyWire DNs shown in *Figure 14* can be viewed at https://flywire.ai/#links/Schlegel2021a/showcase.

Analyses were performed in R and in Python using open source packages. As part of this paper, we have developed various new packages to fetch, process and analyse hemibrain data and integrated them with existing neuroanatomy libraries (*Bates et al., 2020a*). *Table 1* gives an overview of the main software resources used. The packages used for specific analyses will be identified in each section of our methods.

Where appropriate, we have added short tutorials to the documentation of above packages demonstrating some of the analyses performed in this paper. We also provide example code snippets directly related to the analyses in this paper at https://github.com/flyconnectome/2020hemibrain_examples (*Schlegel, 2021*; copy archived at swh:1:rev:73b386e7f6db7a565d2b5eb798c3b06e1d3e0092).

### Neuronal reconstructions in the hemibrain data set

The hemibrain connectome (*Scheffer et al., 2020*) has been largely automatically reconstructed using flood-filling networks (*Januszewski et al., 2018*) from data acquired by focused ion-beam milling scanning EM (FIB-SEM) (*Knott et al., 2008*; *Xu et al., 2017*), followed by manual proofreading. Pre- (T-bars) and postsynapses were identified completely automatically. Significantly, the dense labelling allows estimating completion status as fraction of postsynapses successfully mapped to a neuron. For this first iteration of the hemibrain data set, the completion rate varies between 85% and 16% across neuropils. Notably, the LH currently has one of the lowest completion rates with only ~18% of postsynapses connected mapped to a neuron. We have therefore employed focused semi-manual review of identified neurons in the hemibrain for higher-fidelity connectivity comparison (no manual assessment of synapses). The data can be accessed via the neuPrint connectome analysis service (https://neuprint.janelia.org/) (*Clements et al., 2020*). We built additional software tools to pull, process and analyse these data for R (as part of the natverse ecosystem) (*Bates et al., 2020a*) and Python (see *Table 1*). Neurons can be read from neuPrint and processed (e.g. split into axon and dendrite) with the package hemibrainr using the function hemibrain_read_neurons.

### Neuronal reconstructions in the FAFB data set

Unlike the hemibrain, the FAFB image volume comprises an entire female fly brain (*Zheng et al., 2018*). Two public segmentations of FAFB exist from Google (*Li et al., 2019*) and the Seung lab (https://flywire.ai/) (*Dorkenwald et al., 2020*). However, unlike for the hemibrain data set, these segmentations have not yet been proof-read by humans (at least not at scale). To date, most of the neuronal reconstruction in FAFB has been manual, using CATMAID (*Saalfeld et al., 2009*; *Schneider-Mizell et al., 2016*). We estimate that ~7% of the brain's total neuronal cable, and <1% of its connectivity, has been reconstructed in FAFB by a consortium of 27 laboratories worldwide using CATMAID. For data presented in this work, we have combined coarse morphologies extracted and proof-read from the FlyWire and Google segmentation with detailed manual reconstructions and synapse annotation. We have built software tools to pull, process and analyse these data from CATMAID and FlyWire in R (part of the *natverse* ecosystem) and Python.

### Processing of neuron skeletons and synapse data

Raw skeleton and predicted synapse information from the hemibrain project may have a number of associated issues. Synapses, for example, are sometimes assigned to a neuron's soma or cell body fibre; these are incorrect automatic synapse detections. Autapses are often seen, but the majority of these cases are false-positives (the neuPrint web interface filters those by default). A single neuron

may also have multiple skeletons associated with it that need to be connected. In addition, these skeletons are typically not rooted to their base – that is, the soma if available or, in case of truncated neurons without a soma, the severed cell body fibre. A correctly rooted skeleton is important for some forms of analysis, including axon-dendrite splitting (*Schneider-Mizell et al., 2016*).

We wrote custom code to deal with these issues, as well as split neurons into their axon and dendrite. The correct root of a neuron was identified using an interactive pipeline and expert review (hemibrain_somas). We re-rooted all neurons in the data set (hemibrain_reroot), removed incorrect synapses at somata, along cell body fibres and along primary dendrites (hemibrain_remove_bad_synapses), healed split skeletons, employed a graph-theoretic algorithm to split neurons into axon and dendrites (hemibrain_flow_centrality) and implemented interactive pipelines for users to correct erroneous splits and soma placements. This has enabled us to build putative connectivity edge lists including neuron compartment information (hemibrain_extract_synapses). We have made our code and manipulated data available in our R package hemibrainr.

## Matching neurons between data sets

Hemibrain neurons were matched to those from FAFB, as well as light-level reconstructions, for example, hemilineage models, see *Wong et al., 2013*; *Lovick et al., 2013*, stochastic labelling data (*Dolan et al., 2019*) and images of neuron clones (*Yu et al., 2013*; *Ito et al., 2013*) by bridging these data into the same brain space (*Bogovic et al., 2020*; *Bates et al., 2020a*) and then using NBLAST (*Costa et al., 2016*) to calculate neuron-neuron morphology similarity scores.

Neurons were bridged using the R *nat.jrcbrains* package (https://github.com/natverse/nat.jrcbrains) and nat.templatebrains::xform_brain function or the Python package *navis* (navis.xform_brain) in combination with *navis-flybrains* (https://github.com/schlegelp/navis-flybrains), both of which wrap light-EM bridging registrations reported in *Bogovic et al., 2020*. Prior to NBLAST (using *nat.nblast* or *navis*), EM skeletons were scaled to units of microns, arbour was resampled to $1\mu m$ step size and then converted to vector cloud dotprops format with k = 5 neighbours. To ensure that skeletons from the two EM data sets could be fairly compared, we performed certain postprocessing steps such as pruning away terminal twigs of less than $2 - 5\mu m$ (nat::prune_twigs/navis.prune_twigs) or restricting the arbour for all neurons to the hemibrain volume (hemibrainr::hemibrain_cut) (even if tracing existed outside of this volume for FAFB neurons).

For TOON matching, human experts then visually compared potential matches (with function hemibrain_matching) and qualitatively assessed them as 'good', a near-exact match between the two data sets; 'medium', match definitely represents neurons of the same cell type; and 'poor', neurons are probably the same cell type but under-tracing, registration issues or biological variability made the expert uncertain. We have made our matching pipeline code and matches available in our R package hemibrainr. Matches are available in the package hemibrainr as hemibrain_matches.

## Neurotransmitter assignment

We know the transmitter expression of a few hundred olfactory system neurons based mainly on immunohistochemistry results from the literature (*Tanaka et al., 2012b*; *Wilson and Laurent, 2005*; *Liang et al., 2013*; *Lai et al., 2008*; *Dolan et al., 2019*; *Aso et al., 2014a*; *Okada et al., 2009*; *Tanaka et al., 2012a*). To guess at the transmitter expression of related neurons, we hypothesised that if brain neurons share a hemilineage they will share their fast-acting transmitter expression, as has been seen in the adult ventral nerve cord (*Lacin et al., 2019*). If neuron 1 belongs to the same hemilineage as neuron 2, for which there is data to suggest its neurotransmitter expression, neuron 1 is assumed to express the same neurotransmitter.

## Antennal lobe glomeruli

The antennal lobe (AL) is composed of 58 neuropils called glomeruli. Each glomerulus is a region where a specific type of olfactory or thermo/hygrosensory receptor neurons (ALRNs) synapses onto local and projection neurons, ALLNs and ALPNs, respectively. There are seven identified thermo/hygrosensory glomeruli: VP1d, VP1l, VP1m (*Marin et al., 2020*), VP2, VP3 (*Stocker et al., 1990*), VP4 (*Silbering et al., 2011*; *Frank et al., 2017*; *Knecht et al., 2017*) and 51 olfactory glomeruli (*Bates et al., 2020a*).

**Table 2.** Names of posterior glomeruli across data sets and publications, and supporting reference for names used in this study.

| Glomerulus | Hemibrain v1.1 + 1.2 | Bates et al., 2020b | Tanaka et al., 2012b | Yu et al., 2010 | Receptor | Supporting references |
|---|---|---|---|---|---|---|
| VC5 | VC3m | VC3m | VC3m | – | Ir41a | *Silbering et al., 2011; Task et al., 2020; Hussain et al., 2016; Min et al., 2013; Chai et al., 2019* |
| VC3 | VC3l | VC3l | VC3l | VC3 | Or35a | *Couto et al., 2005; Grabe et al., 2016; Silbering et al., 2011; Task et al., 2020; Min et al., 2013* |
| VM6 (v + m + l) | VC5 | VC5 | VC5 | VM6+ VP1 | Rh50/Amt | *Endo et al., 2007; Li et al., 2016; Chai et al., 2019; Vulpe et al., 2021; Task et al., 2020* |

## Truncated glomeruli

Based on a qualitative assessment, a number of glomeruli (DA4l, DA4m, DM5, VA2, VC5, VM1, VM2, VM3) are substantially (>25%) and 11 (D, DA2, DA3, VA1d, VA6, VA7m, VM6, VM4, VM5d, VM5v, VM7d) are partially (<25%) truncated in the hemibrain. The truncation is due to the proximity of these glomeruli to the 'hot-knife' sections and to the boundary line in the imaging sample (medial and anterior antennal lobe regions).

## Renaming posterior AL glomeruli

Our glomerular identification in *Bates et al., 2020b* was principally based on previously reported projection neurons (ALPNs) associated with a single glomerulus (i.e. 'uniglomerular PNs'). Using PNs provides more points of reference (position of dendrites, lineage, axonal projections) than the relative positions of sensory receptor neuron (ALRN) axon terminals. While this approach works for most glomeruli, some of the posterior glomeruli have had conflicting reports in the literature:

- VC3 has been treated as a single glomerulus (e.g. *Yu et al., 2010*) as well as two separate glomeruli, VC3m and VC3l (e.g. *Tanaka et al., 2012a; Laissue et al., 1999*).
- The VM6 PN has been referred to as VC5 (*Tanaka et al., 2012a*) and VM6+ VP1 (*Yu et al., 2010*)

In collaboration with Karen Menuz (University of Connecticut), Darya Task and Chris Potter (John Hopkins University), Veit Grabe and Silke Sachse (Max Planck Institute for Chemical Ecology), we now consolidate these accounts with extant literature on sensory receptors (see also *Table 2*). As a result of this, three glomeruli were renamed compared with hemibrain v1.1/v1.2 and *Bates et al., 2020b*:

- VC3l → VC3
- VC3m → VC5
- VC5 → VM6

ALRNs of the VM6 glomerulus further split into three distinct subpopulations – VM6v, VM6m and VM6l – with different receptors and origins (*Task et al., 2020; Figure 3—figure supplement 2*). This is likely part of the reason for confusion in the past. Because these subpopulations appear to be indiscernible from the perspective of the downstream network (*Figure 3—figure supplement 2*), we decided to refer to the PNs that cover the combination of VM6v, VM6m and VM6l as 'VM6' uPNs. Following this reasoning, we still refer to the antennal lobe as containing 58 glomeruli. Please see *Table 2* for a summary and supporting references.

These corrections affect names ('instances') and types of ALRNs and ALPNs. Changes will be merged into the hemibrain with the release of version 1.3. All neurons can still be unambiguously identified and tracked across versions of the data set via their body IDs.

## AL glomeruli meshes

The boundary between glomeruli can be defined either using presynapses of ALRNs or the corresponding postsynapses of uniglomerular ALPNs (uPNs). Hence, we generated both ALRN- and ALPN-based glomeruli meshes. These are available in the package hemibrainr as hemibrain_al.surf and in the Supplemental data.

In brief, we used the location of synapses (either dendritic postsynapses of identified uPNs or axonal presynapses of ALRNs) to produce a Gaussian kernel density estimate (KDE) for each glomerulus. We then divided the entire AL into isotropic 480 nm voxels and used the KDEs' point density

functions (pdfs) to assign each voxel to its most likely glomerulus. Voxels with a below-threshold probability to belong to any glomerulus (e.g. tracts) were discarded. The voxel data was postprocessed (binary erosion, fill holes) before being converted to meshes using a marching cubes algorithm. All above steps were performed in Python using scipy (https://www.scipy.org) and scikit-learn (https://scikit-learn.org). Sample code can be found at https://github.com/flyconnectome/2020hemibrain_examples. Finally, the meshes were inspected and manually fixed if required using Blender3d (https://www.blender.org). For the ALPN-based glomeruli meshes, we used the location of dendritic postsynapses of all the uPNs – except for glomeruli VP3, VP5, VP1d, VP1l which do not have clear-cut uPNs and where we used the presynaptic locations of corresponding ALRNs. For the ALRN-based meshes, we used locations of ALRN presynapses. Here, VM2 was excluded because of too few RNs identified for this glomerulus. Also note that for the ALRN-based meshes we used the VM6 ORN subtypes to generate separate meshes for VM6v, VM6m and VM6l.

## Cell-type annotation

Annotations are available via neuPrint and as part of our R package hemibrainr. These are available in the package hemibrainr with the function hemibrain_get_meta.

### Antennal lobe receptor neurons

ALRNs (2643) were identified by morphology and by connectivity to projection neurons. Types were named by the glomerulus they innervate. Soma side was assigned to each ALRN from non-truncated glomeruli whenever possible, based on visual inspection of the path of the neurite towards the nerve entry point.

The number of ALRNs in the 39 whole glomeruli is 1680. For eight types (DC3, VA1v, VA3, VA4, VA5, VA7l, VC2, VC4), although the glomeruli are whole, the majority of ALRNs are fragmented, preventing the assignment of a soma side. For VM6 ALRNs, the glomeruli truncation prevented us from assigning every VM6 ALRN to one of the three populations (12 unassigned). For that reason, in certain instances, we still refer to VM6 ALRNs as one group.

Particularly in truncated glomeruli and glomeruli with fragmented ALRNs, there are many smaller and fragmented bodies for which it is not possible to say if they represent a unique ALRN or if they will merge to another body. Although we have tried to identify these fragments, we cannot be sure that the total number of ALRN bodies is an accurate representation of the number of ALRNs.

In addition to the 2644 ALRNs that we were able to classify, there were 10 that presented issues. Two could be identified as ALRNs but their glomerular arbourisation was missing; therefore, a type could not be assigned (IDs 2197880387, 1852093746, not listed in Supplementary File). Three typed ORNs were excluded because they were pending fixes that altered their connectivity (IDs 1951059936, 2071974816, 5812995304). We also found five outlier ORNs with axon terminals not confined to one glomerulus either two glomeruli in one hemisphere, different glomeruli between hemispheres or innervating the antennal lobe hub (IDs 1760080402, 1855835989, 2229278366, 2041285497, 5813071357).

To assess potential subdivisions of ALRN populations within each glomerulus (*Figure 3—figure supplement 2*), we used a modified version of the synapse-based morphological clustering in *Schlegel et al., 2016* coined syNBLAST (implemented in our Python library navis).

### Antennal lobe local neurons

Candidate neurons (4973) were first identified as any neuron that had at least 5% of its pre- or postsynapses in the AL. From these we excluded the already typed ALPNs (338) and ALRNs (2653), resulting in a candidate list (307) of ALLNs. Among these, only 197 could be typed in accordance with their lineage, morphology and connectivity. The remaining 110 ALLNs are too fragmented to classify and were not used further. Only the ALLNs from the right hemisphere (196) were included in the analysis.

Lineages were identified on the basis of soma and cell body fibre location, partially shared with ALPNs. Next, major groups were assigned in accordance with the previously described neurite morphologies (*Chou et al., 2010*). Due to truncated glomeruli in the data set, we decided to not distinguish between ALLNs innervating all but a few glomeruli vs. most glomeruli; thus both groups are classified as broad ALLNs. The 74 cell types were assigned based on the major morphology class, presence/absence of a bilateral projection, glomerular innervation patterns and neurite density. The

ALLN types were named by concatenating lineage, ID number/capital letter combination and a small letter, in case of strong connectivity differences. The first six ID numbers match the previously identified ALLN types in *Tanaka et al., 2012b*, *Tanaka et al., 2012a*; the following are newly identified types, in decreasing order of arbour size.

## Antennal lobe projection neurons

uPNs were identified by morphology and classified according to our recent complete inventory from the FAFB data set by matching neurons with the help of NBLAST (*Bates et al., 2020b*). mPNs have not been comprehensively typed in past studies. Therefore, mPNs types for hemibrain v1.1/v1.2 were determined in coordination with Kei Ito, Masayoshi Ito and Shin-ya Takemura using a combination of morphological and connectivity clustering. These v1.1/v1.2 mPN types were deliberately very fine-grained to facilitate potential changes (e.g. merges) future releases. See also the paragraph on ALPN analyses below.

## Non-MB olfactory third-order neurons

Non-MB olfactory TOONs were defined as neurons downstream of ALPN axons outside of the MB calyx. They must receive 1% of their synaptic input (or else 10 connections) from an olfactory ALPN, or otherwise 10% of their input (or else 100 connections) from any combination of olfactory ALPNs. This search yields 2383 identifiable, and mainly complete, neuron morphologies. TOONs comprise a range of neuron classes, including a small number of second- and third-order neurons of the gustatory, mechanosensory and visual systems, as well as dopaminergic neurons of the MB, DNs to the ventral nervous system and, most prominently, neurons of the LH.

## Lateral horn neurons

LHNs were defined as a subset of TOONs that have at least 10 pre- or postsynapses in the LH volume (as defined in the hemibrain). We named these cells by extending the LHN naming scheme from *Frechter et al., 2019*, except for cell types with more prominent names already in use in the literature. Neurons were first divided into their hemilineages, indicated by the path of their cell body fibres, for example, DPLm2 (*Lovick et al., 2013*). Hemilineage matches were made to both FAFB and light-level data in order to verify their composition. To simplify the naming of neurons, hemilineages and primary neurons (those cells born in the embryo, which do not fasciculate strongly with secondary hemilineages in the adult brain) were grouped into similar-looking groups, for example, PV5 (posterior-ventral to the LH, 5). Next, neurons within each hemilineage were grouped into coarse morphological sets, termed 'anatomy groups', for example, PV5a. Within each anatomy group, LHNs were broken into morphological cell types using NBLAST, followed by manual curation, for example, PV5a1. Partial reconstructions in FAFB, concatenated using automatically reconstructed neuron fragments (*Li et al., 2019*), were used to help resolve edge cases, that is, by examining which morphological variations appeared consistent between data sets. Neurons were further subdivided into connectivity types (i.e. 'cell type_letter') using CBLAST (*Scheffer et al., 2020*), for example, LHPV5a1_a. With so many new types being added, our expansion of the *Frechter et al., 2019* LH naming system incurred some changes. We have tried to keep names used in main sequence figures in our previous publications (*Dolan et al., 2019*; *Frechter et al., 2019*; *Bates et al., 2020b*), but some have changed as, for example, the hemibrain data has revealed that neurons originate from a different hemilineage or neurons we had once considered to be of the same cell type have different connectivity profiles. Code for these analyses can be found in our R data package, lhns and hemibrainr.

## Descending neurons

The hemibrain v1.1/v1.2 data set includes cell-type information for 109 DNs (*Namiki et al., 2018*), 88 with somata on the right-hand side of the brain. Given that the hemibrain volume does not include the neck connective, ambiguous or previously unknown DNs are difficult to identify. We sought to identify as many DNs as possible without explicitly defining the cell types (many of which are not previously reported in the literature). We used several data sources to help identify DNs including manual and automated tracing in FAFB (*Zheng et al., 2018*; *Li et al., 2019*) and the neuronbridge search tool (https://neuronbridge.janelia.org/, https://github.com/JaneliaSciComp/neuronbridge, *Meissner et al., 2020*; *Otsuna et al., 2018*, also see our R package neuronbridger). The single-most

comprehensive source of information is the recent FlyWire segmentation of the FAFB volume (https://flywire.ai/) (*Dorkenwald et al., 2020*) where we reconstructed neurons that descend from the brain through the neck connective. These FAFB DNs were cross-matched against all hemibrain neurons using NBLAST and subsequent manual curation. This enabled us to identify an additional 236 hemibrain neurons as DNs (see Supplementary Files). A detailed cell typing of these DNs based on combining both data sets will be presented in a future manuscript.

## Graph traversal model

To sort hemibrain neurons into layers with respect to the olfactory system, we employ a simple probabilistic graph traversal model. The model starts with a given pool of neurons – receptor neurons (ALRNs) in our case – as seeds. It then pulls in neurons directly downstream of those neurons already in the pool. This process is repeated until all neurons in the graph have been 'traversed' and we keep track of at which step each neuron was visited. Here, the probability of a not-yet-traversed neuron to be added to the pool depends on the fraction of the inputs it receives from neurons already in the pool. We use a linear function to determine the probability $P_{ij}$ of a traversal from neuron $i$ to $j$:

$$P_{ij} = \begin{cases} \frac{w_{ij}}{(\sum_k w_{kj}) * 0.3} & \text{if} P_{ij} \leq 1 \\ 1 & \text{if} P_{ij} > 1, \end{cases}$$

where $w_{ij}$ is the number of synaptic connections from $i$ to $j$. In simple terms: if the connection from neuron $i$ makes up 30% or more of neuron $j$'s inputs, there is a 100% chance of it being traversed. Each connection from a neuron already in the pool to a neuron outside the pool has an independent chance to be traversed. The threshold of 30% was determined empirically such that known neuron classes like ALLNs and ALPNs are assigned to the intuitively 'correct' layer.

The graph traversal was repeated 10,000 times for the global models (*Figure 2* and *Figure 2— figure supplement 1*) and 5000 per type for the by-RN-type analysis (*Figure 14*). Layers were then produced from the mean across all runs. The code for the traversal model is part of *navis* (https://github.com/schlegelp/navis).

To generate the graph, we used all hemibrain v1.2 neurons with either a type annotation or status label 'Traced' or 'Roughly traced'. We then took the edges between those neurons and removed (a) single-synapse connections to reduce noise and (b) connections between Kenyon cells which are considered false positives (*Li et al., 2020*). This produced a graph encompassing 12.6M chemical synapses across 1.7M edges between 24.6k neurons. Outputs of the model as used in this paper are available in the package hemibrainr as hemibrain_olfactory_layers.

## Class-compartment separation score

This score is inspired by the synapse segregation index used in *Schneider-Mizell et al., 2016*. ALPN innervation of a dendrite is first normalised by the total amount of innervation by ALPNs (*d.pn*) and MBONs (*d.mbon*):

$$d.total = d.mbon + d.pn$$

$$D = d.pn/d.total$$

A dendrite segregation index is then calculated as

$$d.si = -(D * log_{10}(D) - (1 - D) * log_{10}(1 - D))$$

where D is the proportion of dendritic innervation by ALPNs, divided by the total dendritic innervation by MBONs and ALPNs. The axon segregation index (*a.si*) is calculated for the axon of the same neuron. Then the entropy is taken as:

$$e = (1/(d.total + a.total)) * ((a.si * a.total) + (d.si * d.total))$$

$$PN = (d.pn + a.pn)/(d.total + a.total)$$

$$c = -(PN * log_{10}(PN) + (1 - PN))$$

$$segregation.score = 1 - (e/c)$$

## ALRN analyses

ALRN analysis included only those ALRNs for which a glomerular type has been assigned and it excluded glomeruli that are truncated (see 'Antennal lobe glomeruli'). Additionally, any analysis that relied on soma side excluded the eight types that have whole glomeruli but have truncated ALRNs (DC3, VA1v, VA3, VA4, VA5, VA7l, VC2, VC4). Only bilateral ORNs were used for laterality comparisons as only one of seven TRN/HRN types is bilateral.

In connectivity plots, the category 'other' includes any neuron that has been identified, but is not an ALRN, ALPN or ALLN. 'Unknown' refers to unannotated bodies; this might include potential ALRN fragments that cannot be identified.

ALRN presynaptic density was calculated using skeletons and presynapses subsetted to the relevant ALRN-based glomerulus mesh.

## ALPN analyses

### Across-data set morphological clustering

For clustering ALPNs across data sets (hemibrain vs. FAFB right vs. FAFB left), we first transformed their skeletons from their respective template brains to the JRC2018F space. FAFB left ALPNs were additionally mirrored to the right (*Bogovic et al., 2020*; *Bates et al., 2020a*). We then used NBLAST to produce morphological similarity scores between ALPNs of the same (hemi-)lineage (*Costa et al., 2016*). For NBLASTs between hemibrain and FAFB ALPNs, the FAFB ALPNs were first pruned to the hemibrain volume such that they were similarly truncated. The pairwise NBLAST scores were generated from the minimum between the forward (query → target) and reverse (query ← target) scores.

Next, we used the NBLAST scores to – for each ALPN – find the best matches among the ALPNs in the other two data sets. Conceptually, unique ALPNs should exhibit a clear 1:1:1 matching where the best across-data set match is always reciprocal. For ALPN types with multiple representatives, we expect that individuals cannot be tracked across data set because matches are not necessarily reciprocal. We used a graph representation of this network of top matches to produce clusters (*Figure 6—figure supplement 1B*). These initial clusters still contained incorrect merges due to a small number of 'pathological' ALPNs (e.g. from developmental aberrations) which introduce incorrect edges to the graph. To compensate for such cases, we used all pairwise scores (not just the top NBLAST scores) to refine the clusters by finding the minimal cut(s) required to break clusters such that the worst within-cluster score was ≥0.4 (*Figure 6—figure supplement 1C*). This value was determined empirically using the known uPN types as landmarks. Without additional manual intervention, this approach correctly reproduced all 'canonical' (i.e. repeatedly described across multiple studies) uPN types. We note though that in some cases this unsupervised clustering still requires manual curation. We point out some exemplary cases in *Figure 6—figure supplement 1F–J*. For example, M_adPNm4's exhibit features of uniglomerular VC3 adPNs and as a result are incorrectly co-clustered with them. Likewise, a single VC5 lvPN invades the VM4 glomerulus and is therefore co-clustered with the already rather similar looking VM4 lvPNs. In such cases, connectivity information could potentially be used to inform the refinement of the initial clusters.

### Connectivity

Analyses of ALPN connectivity excluded glomeruli that are truncated (see 'Antennal lobe glomeruli'). Additionally, any analysis that relied on ALRN soma side (i.e. ipsilateral ALRNs vs. contralateral) excluded the eight glomeruli that are whole but have truncated ALRNs (DC3, VA1v, VA3, VA4, VA5, VA7l, VC2, VC4). In connectivity plots, the category 'other' includes any neuron that has been identified but is not an ALRN, ALPN or ALLN. 'Unknown' refers to unannotated bodies; this might include potential RN fragments that cannot be identified.

## ALLN analyses

The main theme of the ALLN analysis is to quantify the differences across ALLN types (based on morphology) in innervation (synapses across glomeruli, co-innervation, intra-glomerular morphology) and connectivity motifs. For all of the ALLN analyses, glomerular meshes based on the ALPN-based glomeruli were used.

### Synaptic distribution across glomeruli

The main goal of this analysis was to understand how synapses are distributed across the glomeruli, for the ALLN types. First, for each morphological type, we constructed a matrix with columns representing neurons and rows representing glomeruli. Each element in this matrix has the number of synapses of the specific neuron in the corresponding glomerulus. Synapses per neuron were fetched using the *neuprint-python* package (Python, https://github.com/connectome-neuprint/neuprint-python). Second, for each neuron, glomerular identities were collapsed and sorted by descending order. Third, each column (neuron) was normalised from a range of 0–1 using the minmax scaler from the *scikit-learn* (Python, https://scikit-learn.org/) package. Fourth, the cumulative sum per column was computed. The resulting matrix is composed of each column (neuron) where synaptic score is ordered in a cumulative way.

### Glomerular co-innervation

The main goal of this analysis was to identify pairs of glomeruli that are strongly co-innervated by different ALLN types. For defining co-innervation, the number of synapses in the specific glomeruli from the specific neuron would be used. First, for each morphological type, we constructed a matrix where columns represented neurons and rows represented glomeruli. Each element in this matrix reflected the number of synapses of that neuron in that specific glomerulus. Synapses per neuron were fetched using the *neuprint-python* package. Second, the possible combinations of pairs of glomeruli (that are un-cut) were computed: 39 C_2 or 741 total pairs. Third, for each combination pair the synapses that are co-occurring within a neuron were calculated, resulting in a matrix of dimensions combination pairs (741) by number of neurons of specific ALLN type. Fourth, co-occurring synapses per pair were summed, resulting in a vector of length combinations. This represented the ground truth of co-occurring synapses. Fifth, after computing the matrix from step 3, we shuffled every row independently (i.e. choosing a neuron and shuffling across the pairs of glomeruli). Sixth, we then performed step 4 with this shuffled matrix and repeated steps 5 and 6 for 20k times. This output represented the shuffled synapses. Seventh, for each pair of glomeruli, we computed the proportion of shuffled synapses (within a specific pair of glomeruli) that are higher than the ground truth; this conveys the likelihood of the ground truth being non-random and hence it is the uncorrected p-value. Lastly, we corrected the p-value for multiple comparisons using the package *statsmodels* (Python, https://www.statsmodels.org/), using the *holm-sidak* procedure with a family-wise error rate of 0.05. The pairs with significant p-values following the correction represent the pairs of glomeruli that are strongly co-innervated by the specific ALLN type.

### Connectivity

The main goal of this analysis was to identify how different ALLN types are connected to olfactory ALRNs, uPNs, mPNs and thermo/hygrohygrosensory ALPNs. The input and output synapses between ALLNs and other categories were fetched using the *neuprint-python* package. ALLNs were categorised into a combination of morphological type (sparse, etc.) and lineage type (v, etc.).

### Intra-glomerular morphology

The main goal of this analysis was to identify how intra-glomerular innervation patterns vary across different ALLN types. First, taking each whole glomerulus in turn, we pruned the arbours for each ALLN within that glomerulus using the *navis* package (Python, https://github.com/schlegelp/navis/). From the pruned ALLNs, we excluded any with less than 80 μm of cable length. Second, we calculated the distance between all pairs of ALLNs within that specific glomerulus. This was done as follows: first, for each ALLN pair, for each node we took the five nearest nodes in the opposite ALLN using the KDTree from the *scipy* package (Python, https://www.scipy.org/) and further computed the mean distance. Second, the same procedure was then repeated for all nodes on both sets of ALLNs, producing mean

**Table 3.** Description of neuron metadata listed in supplementary files.

| Column name | Description |
|---|---|
| bodyid | A unique identifier for a single hemibrain neuron |
| pre | The number of presynapses (outputs) a neuron contains, each of these is polyadic |
| post | The number of postsynapses (inputs) to the neuron |
| upstream | The number of incoming connections to a neuron |
| downstream | The number of outgoing connections from a neuron |
| voxels | Neuron size in voxels |
| soma | Whether the neuron has a soma in the hemibrain volume |
| name | The name of this neuron, as read from neuPrint |
| side | Which brain hemisphere contains the neuron's soma |
| connectivity.type | A subset of neurons within a cell type that share similar connectivity, a connectivity type is distinguished from a cell type by an ending _letter unless there is only one connectivity type for the cell type, defined using CBLAST (*Scheffer et al., 2020*) |
| cell.type | Neurons of a shared morphology that take the same cell body fibre tract and come from the same hemilineage (*Bates et al., 2019*) |
| class | The greater anatomical group to which a neuron belongs, see *Figure 1* |
| cellBodyFiber | The cell body fibre for a neuron, as read from neuPrint (*Scheffer et al., 2020*) |
| ItoLee_Hemilineage | The hemilineage that we reckon this cell type belongs to, based on expert review of light-level data from the K. Ito and T. Lee groups (*Yu et al., 2013*; *Ito et al., 2013*) |
| Hartenstein_Hemilineage | The hemilineage that we reckon this cell type belongs to, based on expert review of light-level data from the V. Hartenstein group (*Wong et al., 2013*; *Lovick et al., 2013*) |
| putative.classic.transmitter | Putative neurotransmitter based on what neurons in the hemilineage in question have been shown to express, out of acetylcholine, GABA and/or glutamate |
| putative.other.transmitter | Potential second neurotransmitter |
| FAFB.match | The ID of the manual match from the FAFB data set, ID indicates a neuron reconstructed in FAFBv14 CATMAID, many of these neurons will be available through Virtual Fly Brain, https://v2.virtualflybrain.org/ |
| FAFB.match.quality | The matcher makers' qualitative assessment of how good this match is: a poor match could be a neuron from a very similar cell type or a highly untraced neuron that may be the correct cell type; an okay match should be a neuron that looks to be from the same morphological cell type but there may be some discrepancies in its arbour; a good match is a neuron that corresponds well between FAFB and the hemibrain data |
| layer | Probabilistic mean path length to neuron from ALRNs, depends on connection strengths |
| layer.ct | The mean layer for cell type, rounded to the nearest whole number |
| axon.outputs | Number of outgoing connections from the neuron's predicted axon |
| dend.outputs | Number of outgoing connections from the neuron's predicted dendrite |
| axon.inputs | Number of incoming connections from the neuron's predicted axon |
| dend.inputs | Number of incoming connections from the neuron's predicted dendrite |
| total.length | Total cable length of the neuron in micrometres |
| axon.length | Total axon cable length of the neuron in micrometres |
| dend.length | Total dendrite cable length of the neuron in micrometres |
| pd.length | Total cable length of the primary dendrite 'linker' between axon and dendrite |
| segregation_index | A quantification of how polarised a neuron is, in terms of its segregation of inputs onto its predicted dendrite and outputs onto its axon, where 0 is no-polarisation and 1 is totally polarised (*Schneider-Mizell et al., 2016*) |
| notes | Other notes from annotators |

distances per node per ALLN. Lastly, we collapsed the IDs of the neurons and computed the mean of the top 10% (largest) of the mean distances. This was considered to be the mean intra-glomerular distance between the ALLNs for that specific glomerulus.

### Input-output segregation

The main goals of this analysis were (1) to identify how different ALLN morphological classes vary in the amount of synaptic input and output across different glomeruli and (2) to compare the same with uPNs and ALRNs. First, for each type, we constructed a presynaptic matrix where columns represented neurons and rows represented glomeruli. Each element in this matrix reflected the number of presynaptic connectors of that neuron in that specific glomerulus. Connectors per neuron were fetched using the *neuprint-python* package. Similarly, we constructed a postsynaptic matrix, where each element reflected the number of postsynapses of that neuron in that specific glomerulus. Second, we performed postprocessing on both the presynaptic and postsynaptic matrix. For each neuron, we ranked glomeruli in descending order by synapse number and then removed those glomeruli accounting for the bottom 5% of the synapses. Third, we computed the difference (input-output segregation) by subtracting presynaptic connectors from the postsynapses per neuron. Here we ignored glomeruli where both presynaptic connectors and postsynapses are zero. Fourth, we collapsed the glomerular identities and sorted all neurons by the difference (input-output segregation). Finally, we computed the mean across the neurons. We gave positive ranks to values above 0 (more input) and negative ranks to values below 0 (more output).

### Clustering of ALLNs by the ratio of their axonal output or dendritic input per glomerulus

The main goal of this analysis (*Figure 4—figure supplement 1G*) was to identify how different ALLN types are polarised across different glomeruli (axon-dendrite split developed using the algorithm from *Schneider-Mizell et al., 2016*). First, we selected only those ALLNs (76) that have a axo-dendritic segregation index of >0.1, that is, they are polarised. Second, for each ALLN we computed the axon and dendritic compartment using the flow-centrality algorithm developed in *Schneider-Mizell et al., 2016*. Third, for each glomerulus and for each ALLN we computed the fraction of dendritic inputs (input synapses located in the dendritic compartment inside the specific glomerulus) to the total dendritic inputs (input synapses located in the dendritic compartment across all glomeruli) and fraction of axonic outputs (output synapses located in the axonic compartment inside the specific glomerulus) to the total axonic outputs (output synapses located in the axonic compartment across all glomeruli). Fourth, we computed a score defined by the fraction of axonic output – the fraction of dendritic input. The higher the score, the greater the ALLN's bias for axonically outputting in a glomerulus, over receiving dendritic input. Fifth, we computed the mean of these scores for different ALLN types across the different glomeruli. Finally, we applied the clustering algorithm (using hierarchical clustering based on Ward's distance using functions from base R) to these scores.

## Supplemental data

We have made our code, with examples, and detailed data available in our R package hemibrainr. Here we provide core data. Please see *Table 3* for a description of the metadata contained in the supplementary files.

## Acknowledgements

This work was supported by a Wellcome Trust Collaborative Award (203261/Z/16/Z) to GSXEJ and GMR; an ERC Consolidator grant (649111) and core support from the MRC (MC-U105188491) to GSXEJ; NIH BRAIN Initiative grant 1RF1MH120679-01 to Davi Bock and GSXEJ; a Boehringer Ingelheim Fonds PhD Fellowship and a Herchel Smith Studentship to ASB; NIH R01DC008174 (to Rachel Wilson) and an F31 fellowship (DC016196) to AB-M; and by the Howard Hughes Medical Institute. We thank the FlyEM team and their collaborators for pre-publication access to the hemibrain data set and reconstructions. Development and administration of the FAFB tracing environment, analysis tools and GSXEJ, TS and PS were funded in part by NIH BRAIN Initiative grant 1RF1MH120679-01 to Davi Bock and GSXEJ, with software development effort and administrative support provided by

Tom Kazimiers (Kazmos GmbH) and Eric Perlman (Yikes LLC). We are also grateful to the Seung and Murthy labs for access to the flywire.ai reconstruction community. We thank Kei Ito, Masayoshi Ito and Shin-ya Takemura for the examination and discussion of neuron types and names in the antennal lobe and lateral horn, and Rachel Wilson for those in the antennal lobe. We also thank Romain Franconville for his contributions to our R package *neuprintr*. We thank Karen Menuz, Darya Task, Veit Grabe, Chris Potter and Silke Sachse for discussions on posterior antennal lobe glomeruli and receptor neuron identity. Finally, we thank Liqun Luo, Kei Ito, Rachel Wilson, Thomas Riemensperger and Andrew Lin for comments on the manuscript.

## Additional information

### Funding

| Funder | Grant reference number | Author |
| --- | --- | --- |
| Wellcome Trust | Collaborative Award 203261/Z/16/Z | Philipp Schlegel<br>Tomke Stürner<br>Sridhar R Jagannathan<br>Nikolas Drummond<br>Joseph Hsu<br>Laia Serratosa Capdevila<br>Alexandre Javier<br>Elizabeth C Marin<br>Imaan FM Tamimi<br>Feng Li<br>Gerald M Rubin<br>Marta Costa<br>Gregory SXE Jefferis |
| European Research Council | Consolidator grant 649111 | Laia Serratosa Capdevila<br>Alexandre Javier<br>Gregory SXE Jefferis |
| Medical Research Council | Core support MC-U105188491 | Alexander Shakeel Bates<br>Gregory SXE Jefferis |
| National Institutes of Health | BRAIN Initiative grant 1RF1MH120679-01 | Philipp Schlegel<br>Tomke Stürner<br>Gregory SXE Jefferis |
| National Institutes of Health | F31 fellowship DC016196 | Asa Barth-Maron |
| Boehringer Ingelheim Fonds | PhD Fellowship | Alexander Shakeel Bates |
| Herchel Smith | Studentship | Alexander Shakeel Bates |
| National Institutes of Health | R01DC008174 | Asa Barth-Maron |
| Howard Hughes Medical Institute | | Gerald M Rubin<br>Stephen M Plaza |

The funders had no role in study design, data collection and interpretation, or the decision to submit the work for publication.

### Author contributions

Philipp Schlegel, Alexander Shakeel Bates, Conceptualization, Data curation, Formal analysis, Investigation, Methodology, Software, Supervision, Validation, Visualization, Writing – original draft, Writing – review and editing; Tomke Stürner, Sridhar R Jagannathan, Conceptualization, Data curation, Formal analysis, Investigation, Methodology, Validation, Visualization, Writing – original draft, Writing – review and editing; Nikolas Drummond, Joseph Hsu, Laia Serratosa Capdevila, Imaan FM Tamimi, Data curation; Alexandre Javier, Data curation, Validation; Elizabeth C Marin, Asa Barth-Maron, Data curation, Validation, Writing – review and editing; Feng Li, Data curation, Supervision; Gerald M Rubin, Funding acquisition, Project administration; Stephen M Plaza, Project administration, resources; Marta Costa,

Conceptualization, Data curation, Formal analysis, Funding acquisition, Investigation, Methodology, Project administration, Supervision, Validation, Visualization, Writing – original draft; Gregory S X E Jefferis, Conceptualization, Data curation, Formal analysis, Funding acquisition, Investigation, Methodology, Project administration, Software, Supervision, Validation, Visualization, Writing – original draft, Writing – review and editing

## Author ORCIDs
Philipp Schlegel (ID) http://orcid.org/0000-0002-5633-1314
Alexander Shakeel Bates (ID) http://orcid.org/0000-0002-1195-0445
Tomke Stürner (ID) http://orcid.org/0000-0003-4054-0784
Sridhar R Jagannathan (ID) http://orcid.org/0000-0002-2078-1145
Elizabeth C Marin (ID) http://orcid.org/0000-0001-6333-0072
Feng Li (ID) http://orcid.org/0000-0002-6658-9175
Gerald M Rubin (ID) http://orcid.org/0000-0001-8762-8703
Stephen M Plaza (ID) http://orcid.org/0000-0001-7425-8555
Marta Costa (ID) http://orcid.org/0000-0001-5948-3092
Gregory S X E Jefferis (ID) http://orcid.org/0000-0002-0587-9355

## Decision letter and Author response
Decision letter https://doi.org/10.7554/eLife.66018.sa1
Author response https://doi.org/10.7554/eLife.66018.sa2

---

# Additional files

## Supplementary files
• Supplementary file 1. Layers assigned by the probabilistic graph traversal model. bodyId refers to neurons' unique ID in neuPrint. layer_mean contains the mean layer after 10,000 iterations of the main model (*Figure 2*). layer_olf_mean and layer_th_mean contain the mean layers from running the traversal model with ORNs and THN/HRNs, respectively (*Figure 2—figure supplement 2*).

• Supplementary file 2. Sensory meta-information related to each glomerulus. Columns: glomerulus (canonical name for one of the 51 olfactory + 7 thermo/hygrosensory antennal lobe glomeruli), laterality (whether the glomerulus receives bilateral or only unilateral innervation from ALRNs), expected_cit (a citation that describes the expected number of RNs in this glomerulus), expected_RN_female_1 h (number of expected RNs in one hemisphere), expected_RN_female_SD (standard deviation in the expected number of RNs), missing (qualitative assessment of glomeruli truncation), RN_frag (if the RNs in that glomerulus are fragmented), receptor (the OR or IR expressed by cognate ALRNs, *Bates et al., 2020b*; *Task et al., 2020*), odour_scenes the general 'odour scene(s)' which this glomerulus may help signal, (*Mansourian and Stensmyr, 2015*; *Bates et al., 2020b*), key_ligand (the ligand that excites the cognate ALRN or receptor the most, based on pooled data from multiple studies, MÃ¼nch and *MÃ¼nch and Galizia, 2016*), valence (the presumed valence of this odour channel, *Badel et al., 2016*). Exists as hemibrain_glomeruli_summary in our R package hemibrainr.

• Supplementary file 3. File listing all identified antennal lobe receptor neurons (ALRNs) in the hemibrain, including information shown in neuPrint. See above for column explanations. Exists as rn. info in our R package hemibrainr.

• Supplementary file 4. All the hemibrain neurons we have classed as antennal lobe local neurons (ALLNs). See above for column explanations. Exists as alln.info in our R package hemibrainr.

• Supplementary file 5. All the hemibrain neurons we have classed as antennal lobe projection neurons (ALPNs). See above for column explanations. In addition, across_dataset_cluster refers to the clustering with left and right FAFB PNs; is_canonical indicates whether that ALPN is one of the well studied "canonical" uPNs. Exists as pn.info in our R package hemibrainr.

• Supplementary file 6. All the hemibrain neurons we have classed as third-order olfactory neurons (TOONs) including lateral horn neurons (LHNs), as well as wedge projection neurons (WEDPNs), lateral horn centrifugal neurons (LHCENT) and other projection neuron classes (*Figure 1*). See above for column explanations. Exists as ton.info in our R package hemibrainr.

• Supplementary file 7. All the hemibrain neurons we have classed as neurons that descend to the ventral nervous system (DNs). See above for column explanations. Exists as dn.info in our R package hemibrainr.

• Supplementary file 8. The root point in hemibrain voxel space, for each hemibrain neuron. This is either the location of the soma, or the tip of a severed cell body fibre tract, where possible. Exists as hemibrain_somas in our R package hemibrainr.

• Supplementary file 9. The start points for different neuron compartments. Nodes downstream of this position in the 3D structure of the neuron indicated with bodyid, belong to the compartment type designated by Label. A product of running flow_centrality on hemibrain neurons, exists as hemibrain_splitpoints in our R package hemibrainr.

• Supplementary file 10. 3D triangle mesh for the hemibrain surface as a.obj file. This mesh was generated by first merging individual ROI meshes from neuPrint and then filling the gaps in between in a semi-manual process. It also exists as hemibrain.surf in our R package hemibrainr.

• Supplementary file 11. 3D meshes of 51 olfactory + 7 thermo/hygrosensory antennal lobe glomeruli for the hemibrain volume, generated from ALRN presynapses. These meshes follow the subdivision of VM6 and hence contain 60 meshes in total. Note that hemibrain coordinate system has the anterior-posterior axis aligned with the Y axis (rather than the Z axis, which is more commonly observed).

• Supplementary file 12. 3D meshes of 51 olfactory + 7 thermo/hygrosensory antennal lobe glomeruli for the hemibrain volume, generated from ALPN postsynapses. Note that hemibrain coordinate system has the anterior-posterior axis aligned with the Y axis (rather than the Z axis, which is more commonly observed). These meshes are also available as hemibrain_al.surf in our R package hemibrainr.

• Transparent reporting form

### Data availability
The hemibrain connectome including our annotations is hosted via neuPrint at https://neuprint.janelia.org. Published data (neuronal reconstructions and connectivity) from the FAFB EM data set is hosted by Virtual Fly Brain (VFB) at https://catmaid.virtualflybrain.org. A snapshot of the FAFB data used in this study will be shared with VFB prior to publication. Meta data (e.g. neuron classifications, axon-dendrite splits, glomeruli meshes, etc) are included in the manuscript and supporting files. In addition, we maintain Github repositories with meta data (https://github.com/flyconnectome/hemibrain_olf_data, copy archived at https://archive.softwareheritage.org/swh:1:rev:dcad3064b258bb807222c31d739c1c917ed309f6) and code examples (https://github.com/flyconnectome/2020hemibrain_examples copy archived at https://archive.softwareheritage.org/swh:1:rev:73b386e7f6db7a565d2b5eb798c3b06e1d3e0092). FlyWire DNs shown in Figure 14 can be viewed at https://flywire.ai/#links/Schlegel2021a/showcase.

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
