## [Decision Letter]

Our editorial process produces two outputs: (i) public reviews designed to be posted alongside the preprint for the benefit of readers; (ii) feedback on the manuscript for the authors, including requests for revisions, shown below.

Thank you for submitting your article "Information flow, cell types and stereotypy in a full olfactory connectome" for consideration by *eLife*. Your article has been reviewed by 3 peer reviewers, and the evaluation has been overseen by Leslie Griffith as Reviewing Editor and Catherine Dulac as the Senior Editor. The following individual involved in review of your submission has agreed to reveal their identity: Liqun Luo (Reviewer #2).

The reviewers have discussed their reviews with one another, and the Reviewing Editor has drafted this to help you prepare a revised submission. Essential revisions:

The reviewers were unanimous in saying that this study requires no new data or analyses. They have provided comments in the reviews below that they feel will help clarify particular points and make the study more accessible.

*Reviewer #1 (Recommendations for the authors):*

I have a few minor suggestions:

1. It would have been interesting to hear more about putative differences or evolving complexity between the larval and adult olfactory systems. The authors compare thermo- and hygrosensation to olfaction, but systematic comparison of larval and adult olfaction might reveal perhaps even more interesting differences that could perhaps be explained by the different living environments and challenges these distinct life stages face.

2. The authors write that the mushroom body is for learning, the lateral horn for innate behavior. They then rightly point out that this is not always true and cite their own work (Dolan et al., 2018). Additional work should be cited here that showed significantly earlier that this distinction is not always true for olfaction as well as thermosensation.

3. With all we know (also from the vertebrate system, e.g. visual system, LGN is plastic), is it still helpful to assign a function to a single light-microscopy level anatomically defined brain region, such as the mushroom body, rather than to a circuit that spans most of the brain and many brain regions? Or alternatively, it might be best to name the synaptic connections between neurons that can store long-term information vs. the ones that cannot. However, since we don't have this information yet for all synapses, we should, in my opinion, step away from these oversimplifications that divide a brain in regions of distinct function rather than describe brain-wide networks. The connectome data is ideal to revisit these preconceptions and to stress that it's best to remain open minded, because as we have seen from such data, it holds surprises such as an unexpected structure of KC innervation.

*Reviewer #2 (Recommendations for the authors):*

I am highly enthusiastic in supporting the publication of this manuscript in *eLife*.

A few specific critiques (all minor) for the authors to address during their revision:

1. lines 94-96: I find the statement "The traversal is probabilistic: the likelihood of a new neuron being added to the pool increases linearly with the fraction of its inputs coming from neurons already in the pool with a threshold at 30% (Figure 2C)." difficult to parse. So is Figure 2C right panel and its very brief legend. Given the importance of this concept, please explain this in a manner that is easier to understand.

2. Line 287: delete "is" from "this is has already".

3. Figure 8A: The composition "TOONs" is a bit unclear. The inner ring contains LHN, "other TOON", WED PN, and many unlabeled slices. Are these all TOONs? According to the legend, these unlabeled slices include DANs, MBONs, and so on. Are these all directly postsynaptic to ALPNs according to definition of TOONs?

*Reviewer #3 (Recommendations for the authors):*

Figure 4D. It would help to clarify this comparison, perhaps by adding an arrowhead to the right panel.

Figure 13C, D "% of TONs" typo.

Line 560. "Figure 5D"; this is a bit confusing following the reference to il3LN6.

Line 567 "Figure 5F" 5G?

Line 467 "Figure 13—figure supplement 1G" G is a typo.

Line 467 Figure 13—figure supplement 1. It would be helpful to indicate more clearly the three main groups that are said to emerge.

Figure 6 – Figure Supp 1C – please define the red circles.

---

## [Author Response]

Essential revisions:The reviewers were unanimous in saying that this study requires no new data or analyses. They have provided comments in the reviews below that they feel will help clarify particular points and make the study more accessible.Reviewer #1 (Recommendations for the authors):I have a few minor suggestions:1. It would have been interesting to hear more about putative differences or evolving complexity between the larval and adult olfactory systems. The authors compare thermo- and hygrosensation to olfaction, but systematic comparison of larval and adult olfaction might reveal perhaps even more interesting differences that could perhaps be explained by the different living environments and challenges these distinct life stages face.

We previously provided a numerical comparison of the early olfactory system in larval and adult *Drosophila* that illustrates the increase in complexity in the adult olfactory system (Bates, Schlegel et al., 2020). We agree that a connectivity-based comparison would indeed be very interesting. However, many motifs that we already analyse do not exist in the larva (e.g. ALRN lateralisation) or lack publicly available data (e.g. third order olfactory neurons for the larva). Therefore, any new and meaningful insight into larval vs adult olfaction would require substantial new analyses orthogonal to those already presented and we feel that this is unfortunately not within the scope of this paper. In addition, the full larval connectome is not yet published and publicly available though it is expected soon (Winding et al. in prep, Cardona group). Therefore there should be opportunities to address these questions properly in the future.

2. The authors write that the mushroom body is for learning, the lateral horn for innate behavior. They then rightly point out that this is not always true and cite their own work (Dolan et al., 2018). Additional work should be cited here that showed significantly earlier that this distinction is not always true for olfaction as well as thermosensation.

We have addressed this comment by adding a number of additional citations (primary research papers and reviews) to both the introduction and the results paragraph on “Higher-order olfactory neurons”. These include e.g. Heisenberg 2003, Yu et al. 2004, Sejourne et al. 2011, Zhao et al. 2019 and Kadow 2019.

3. With all we know (also from the vertebrate system, e.g. visual system, LGN is plastic), is it still helpful to assign a function to a single light-microscopy level anatomically defined brain region, such as the mushroom body, rather than to a circuit that spans most of the brain and many brain regions? Or alternatively, it might be best to name the synaptic connections between neurons that can store long-term information vs. the ones that cannot. However, since we don't have this information yet for all synapses, we should, in my opinion, step away from these oversimplifications that divide a brain in regions of distinct function rather than describe brain-wide networks. The connectome data is ideal to revisit these preconceptions and to stress that it's best to remain open minded, because as we have seen from such data, it holds surprises such as an unexpected structure of KC innervation.

We have adjusted our introduction and the results paragraph on “Higher-order olfactory neurons” to de-emphasise structure-function relationships.

Reviewer #2 (Recommendations for the authors):I am highly enthusiastic in supporting the publication of this manuscript in eLife.A few specific critiques (all minor) for the authors to address during their revision:1. lines 94-96: I find the statement "The traversal is probabilistic: the likelihood of a new neuron being added to the pool increases linearly with the fraction of its inputs coming from neurons already in the pool with a threshold at 30% (Figure 2C)." difficult to parse. So is Figure 2C right panel and its very brief legend. Given the importance of this concept, please explain this in a manner that is easier to understand.

We have updated this section of the results (“Layers in the olfactory system”) to address these comments. In particular, we rephrased the explanation of the traversal model to make it clearer and now provide a more explicit example for its probabilistic nature. In addition, we mention some of the caveats of the model such as not taking the sign (excitatory vs inhibitory) of a connection into account.

2. Line 287: delete "is" from "this is has already".3. Figure 8A: The composition "TOONs" is a bit unclear. The inner ring contains LHN, "other TOON", WED PN, and many unlabeled slices. Are these all TOONs? According to the legend, these unlabeled slices include DANs, MBONs, and so on. Are these all directly postsynaptic to ALPNs according to definition of TOONs?

To address this comment, we have updated the legend of figure 8 and the corresponding paragraph (“Higher-order olfactory neurons”) and now provide an explicit definition of “TOON”. In brief, third-order olfactory neurons have historically (i.e. in absence of a connectome) been defined by overlap with olfactory projection neurons (ALPNs) and we have translated this definition to “any neuron that receives >= 1% of its inputs from ALPNs outside of the MB”. This does include a small number of neurons in other classes, such as DANs and WEDPNs (see also the supplementary data).

Reviewer #3 (Recommendations for the authors):Figure 4D. It would help to clarify this comparison, perhaps by adding an arrowhead to the right panel.Figure 13C, D "% of TONs" typo.Line 560. "Figure 5D"; this is a bit confusing following the reference to il3LN6.Line 567 "Figure 5F" 5G?Line 467 "Figure 13—figure supplement 1G" G is a typo.Line 467 Figure 13—figure supplement 1. It would be helpful to indicate more clearly the three main groups that are said to emerge.Figure 6 – Figure Supp 1C – please define the red circles.

All fixed, including small updates to Figure 4D (added arrowheads), Figure 13—figure supplement 1 (added boxes to highlight groups) and Figure 6—figure supplement 1C (explained red circles).